# Defects engineering simultaneously enhances activity and recyclability of MOFs in selective hydrogenation of biomass

Wenlong Xu[1,3], Yuwei Zhang[1,3], Junjun Wang[2,3], Yixiu Xu[1], Li Bian[1], Qiang Ju[1], Yuemin Wang [2✉] & Zhenlan Fang [1✉]

The development of synthetic methodologies towards enhanced performance in biomass conversion is desirable due to the growing energy demand. Here we design two types of Ru impregnated MIL-100-Cr defect engineered metal-organic frameworks (Ru@DEMOFs) by incorporating defective ligands (DLs), aiming at highly efficient catalysts for biomass hydrogenation. Our results show that Ru@DEMOFs simultaneously exhibit boosted recyclability, selectivity and activity with the turnover frequency being about 10 times higher than the reported values of polymer supported Ru towards D-glucose hydrogenation. This work provides in-depth insights into (i) the evolution of various defects in the cationic framework upon DLs incorporation and Ru impregnation, (ii) the special effect of each type of defects on the electron density of Ru nanoparticles and activation of reactants, and (iii) the respective role of defects, confined Ru particles and metal single active sites in the catalytic performance of Ru@DEMOFs for D-glucose selective hydrogenation as well as their synergistic catalytic mechanism.

[1] Key Laboratory of Flexible Electronics (KLOFE) & Institute of Advanced Materials (IAM), Jiangsu National Synergetic Innovation Center for Advanced Materials (SICAM), Nanjing Tech University (NanjingTech), 30 South Puzhu Road, Nanjing 211816, China. [2] Institute of Functional Interfaces, Karlsruhe Institute of Technology (KIT), 76344 Eggenstein-Leopoldshafen, Germany. [3]These authors contributed equally: Wenlong Xu, Yuwei Zhang, Junjun Wang. ✉email: yuemin.wang@kit.edu; iamzlfang@njtech.edu.cn

Biomass such as cellulose and hemicellulose primarily derived from agricultural wastes is a renewable alternative to fossil fuels and a sustainable carbon feedstock for chemical production due to its renewability, low pollution, wide distribution, and abundance[1]. D-glucose monomers as the main feedstock, obtained from the deconstruction of cellulose or hemicellulose, provide a biorefinery platform for the production of sorbitol, mannitol, fuel alcohol, or other high value-added chemicals via chemical catalytic or biocatalytic processes.

In contrast to enzymes and metal oxides, metal−organic frameworks (MOFs)-based heterogeneous catalysts (MOF-HCs) offer wondrous opportunities to transform a wide range of biomass-derived sugar-containing chemicals into targeted products over a liberal operational regulatory process due to their high surface area, pristine metal single active sites (MSAS), well dispersion of loaded active species, tunable functionalities, readily adjustable pore size, and porosity[2–6]. However, MOF-HCs often encounter low reusability under actual reaction conditions, thus hindering their further catalytic applications.

Recently, defect-engineered MOFs (DEMOFs) receive increasing attention along with the development of characterization techniques[7–10] due to the crucial roles of defects in gas storage[11,12], selective adsorption[13], drug delivery[14–16], catalysis[2–4,13,17–21], mechanical response[19], conductivity[22,23], magnetic[24], chirality[25], etc. Defect-engineering strategy is demonstrated as an efficient method to enhance the activity of MOFs due to the modified pore structure and changes in coordination environment and electronic structure of metal coordination unsaturated sites (CUSs)[4,8,17,18,26–32]. However, it remains a great challenge to simultaneously enhance the activity, selectivity, and recyclability of MOF-HCs due to the lower stability of DEMOFs in reaction solutions, resulting from the missing bonds and/or larger cavities at defective sites.

Metal active moieties encapsulated DEMOFs (AM@DEMOFs) catalysts generally show enhanced catalytic performance through synergistic effects[17,33,34]. Ruthenium (Ru) nanoparticles (NPs) are widely used in biomass hydrogenation reduction reactions due to their excellent catalytic efficiency[35]. The nature of metal nodes in the framework has a critical effect on the stability of the framework[36] and the electron density of embedded metal NPs, playing a key role in the activity of AM@DEMOFs[6]. Consequently, we anticipate that a special type of defects at metal nodes should have a unique regulation effect on the framework stability and the morphology and electron density of embedded metal NPs due to the modification of pore structures and interactions between NPs, CUSs, and defective sites[13,17,37], providing a great possibility to simultaneously boost the activity, selectivity and recyclability of AM@DEMOFs catalysts through a rational design and tuning of defects. Furthermore, Cr(III)-based MIL-100 (MIL-100-Cr) possessing a cationic framework is more stable under hydrothermal conditions than Fe(III)-based MIL-100 due to the intrinsically smaller metal−ligand exchange constant of Cr(III) compared with Fe(III)[36].

In this work, based on above considerations together with the aim of producing sorbitol via D-glucose selective hydrogenation (biomass refinement reaction), we design two different types of Ru NPs impregnated MIL-100-Cr DEMOFs (Ru@$D_{xp}$, $x = 1$, 2; $p = a – c$, Fig. 1) via doping 3,5-pyridinedicarboxylate (DL1: PyDC$^{2-}$, type-A defect, Figs. 1b and 2a) and m-phthalate (DL2: $m$-BDC$^{2-}$, type-B defect, Figs. 1c and 2b) with different feeding ratios ($z$) of defective ligand (DL) to total ligands (TLs, a: $z = 10\%$, b: $z = 30\%$, c: $z = 50\%$), respectively. Herein, DL1 and DL2 possess the same structural configuration, being similar to that of parent linker 1,3,5-benzenetricarboxylate (L0: btc$^{3-}$, Figs. 1a and 2c) but missing a carboxylate group. The only difference between type-A and type-B defects is that the

1-substituted position of the aromatic ring of DL1 in Ru@$D_{1P}$ has a ligator N atom (basic pyridyl-N site) to bind to Ru NPs (Fig. 1d), reactants and/or products (Fig. 1f), which is absent in Ru@$D_{2P}$ (Fig. 1e, g). This work addresses several important issues such as tolerance of cationic frameworks to DL$x$, the modification of the electronic and steric properties of MSAS, the compensation of missing charges, the evolution of various defects upon DL$x$ incorporation and Ru impregnation, the unique effect of each type of defects on the electron density of impregnated Ru NPs and the adsorption mode of reactant, the special role of defects, Ru NPs and MSAS on the activity, selectivity and recyclability of these designed Ru@DEMOFs, and their synergistic catalytic mechanism of D-glucose selective hydrogenation to sorbitol (Fig. 1f, g).

## Results and discussion

**Structure and composition determination.** The synthesis details and formula of the parent MIL-100-Cr ($D_0$) and two kinds of DEMOFs ($D_{xp}$: [Cr$_3$F$_y$(H$_2$O)$_3$O(btc)$_{2-z'}$(DL$x$)$_{z'}$]·nH$_2$O and Ru@DEMOFs (Ru@$D_{xp}$: {Ru$_m$[Cr$_3$F$_y$(H$_2$O)$_3$O(btc)$_{2-z'}$(DL$x$)$_{z'}$]·nH$_2$O ($x = 1$ or 2; $p = a – c$) are listed in Supplementary Table 1. The powder X-ray diffraction (PXRD) peaks of all obtained samples are indexed and match well with the simulated XRD pattern of pristine MIL-100-Cr (Supplementary Figs. 1 and 2), demonstrating the phase purity of all samples. The PXRD patterns, together with complementary data obtained by elemental analyzer (EA), thermal gravimetric analysis (TGA), routine Fourier transformed infrared spectroscopy (FTIR), nuclear magnetic resonance spectroscopy (NMR), and inductively coupled plasma optical emission spectroscopy (ICP-OES), rule out the presence of any physiosorbed DLs, thus confirming the DL tolerance of MIL-100-Cr framework (Supplementary Figs. 3–10 and Supplementary Tables 2 and 3). Considering the cationic nature of the framework, we have conducted X-ray photoelectron spectroscopy (XPS) analysis for all samples (Supplementary Figs. 11–13). The XPS results exclude the possibility that the missing negative charges due to the DL incorporation were compensated by increasing the concentration of F⁻ counterions.

The Brunauer–Emmett–Teller (BET) surface areas and pore size distribution of all samples are determined based on N$_2$ adsorption-desorption isotherms (Supplementary Figs. 14–22). The results reveal a disparity in tolerance of the cationic framework to DL$x$ ($x = 1$, 2) and in the defect construction ability of these two DLs. Moreover, it is found that the modification effects of Ru impregnation on the porosity, pore structure, defects, and crystallinity of DEMOFs vary depending on the type and content of DL. In comparison to $D_0$, $D_{1a-b}$ and $D_{2a-b}$ show only a small decrease in BET values with a pronounced increase in mesopores size. The better tolerance of this cationic framework to DL1 is mainly attributed to the weak interaction between pyridyl-N atoms (Lewis base) in DL1 and Cr ions (Lewis acid, Fig. 1b). The BET surface areas of Ru@$D_{1a-b}$ and Ru@$D_{2a-b}$ decrease slightly along with doping of Ru NPs, accompanied by the formation of additional mesopores (Supplementary Table 4). Overall, these results confirm that Ru@DEMOFs (DL doping level $z \leq 30\%$) preserve the structural integrity of MIL-100-Cr, in line with the PXRD, FTIR, and TGA results (Supplementary Sections 1.5–1.7).

**Modulation of impregnated Ru NPs by two types of defects.** To determine the unique effect of each type of defects on regulating the amount, dispersion, size, shape, and stability of Ru NPs, scanning transmission electron microscopy (STEM) (Fig. 2d–i and Supplementary Figs. 23–26) and ICP-OES experiments (Supplementary Table 5) have been conducted on Ru@$D_0$ and

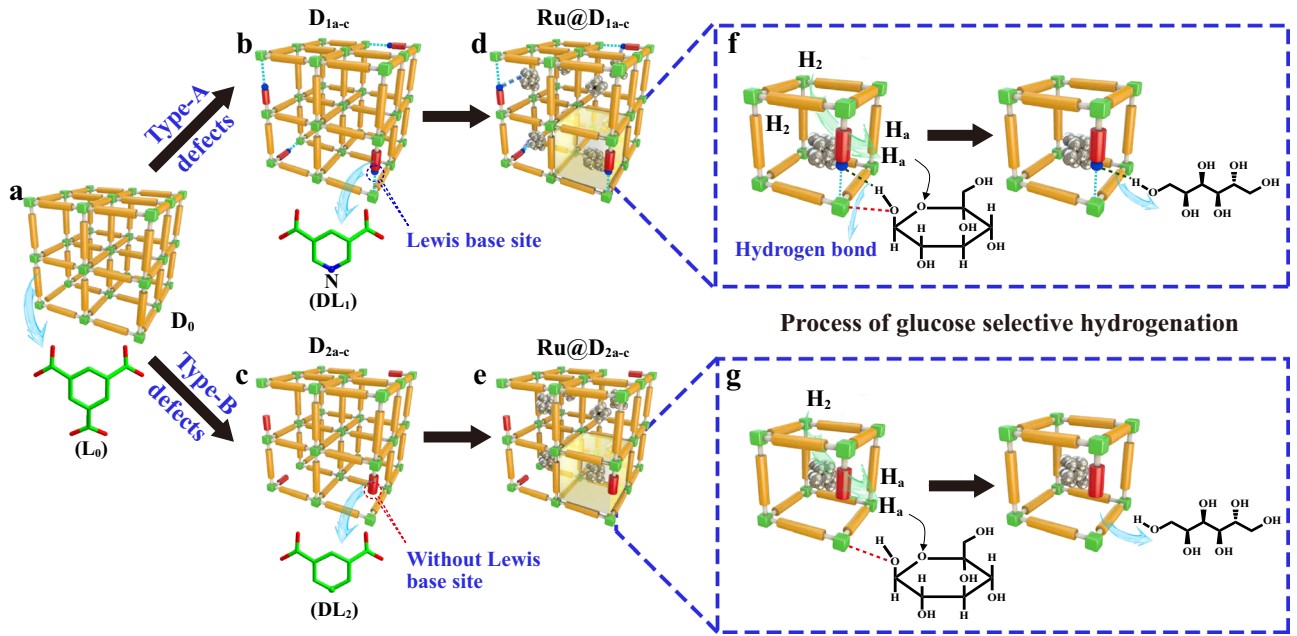

**Fig. 1 Formation of two different types of defects and their corresponding catalytic mechanisms. a–e** Schematic illustration of parent MIL-100-Cr MOF $D_0$ (**a**), $D_{1a-c}$ DEMOFs with type-A defects, generated via incorporation of DL1 with a weak ligator (pyridyl-N) at defect sites (**b**), $D_{2a-c}$ DEMOFs with type-B defects, generated via incorporation of DL2 without ligator at defect sites (**c**), Ru-impregnated D1a-c catalysts (Ru@$D_{1a-c}$) (**d**), and Ru-impregnated $D_{2a-c}$ catalysts (Ru@$D_{2a-c}$) (**e**). **f, g** Proposed D-glucose hydrogenation mechanisms catalyzed by Ru@$D_{1a-c}$ (**f**) and Ru@$D_{2a-c}$ catalysts (**g**). The yellow sticks, the shorter red sticks with a blue ball, and the bare shorter red sticks represent parent L0, DL1, and DL2 ligands, respectively. The green cubes represent metal sites. Both **f** and **g** show the enlarged perspective views of the unit containing defect sites.

Ru@DEMOFs before and after catalyzing selective hydrogenation of D-glucose to sorbitol. The STEM images provide direct evidence for the presence of small confined Ru NPs embedded in DEMOFs, where the size of dominant Ru NPs in Ru@$D_{1a}$ (~1.95 nm, Fig. 2j) and Ru@$D_{2a}$ (~1.93 nm, Fig. 2k) is comparable to that in Ru@$D_0$ (~1.98 nm) (Fig. 2l). The content and size of impregnated Ru NPs in Ru@DEMOFs increase upon increasing the feeding ratio of DL$x$ (Supplementary Tables 5 and 6), which is consistent with the evolution of the modified pore structures of Ru@DEMOFs. Interestingly, the content of impregnated Ru NPs in Ru@$D_0$, Ru@$D_{1c}$, and Ru@$D_{2c}$ is slightly higher than the feeding ratio. This is primarily due to the partial framework decomposition during Ru impregnation, supported by the weight loss of MOFs (Supplementary Table 5) and the leaching $Cr^{3+}$ cations in the ethanol solution of RuCl$_3$ after Ru impregnation. The latter process has a distinguishable corrosion effect on the framework of pristine MIL-100-Cr ($D_0$) as well as $D_{1c}$ and $D_{2c}$ DEMOFs, but a negligible effect on $D_{1a-b}$ and $D_{2a-b}$ DEMOFs with lower DL$x$ concentrations. These results demonstrate that the introduced defects either type-A or type-B, constructed by the incorporation of DL$x$ ($x = 1, 2$) with $z \leq 30\%$, can enhance the framework stability of MIL-100-Cr. This finding is probably attributed to the coordination modulation effect of DL$x$, acting as coligand, on the crystal growth of MIL-100-Cr DEMOFs at the molecular level during in situ synthesis[18,32].

The results of CO chemisorption reveal that the dispersion of confined Ru NPs in Ru@DEMOFs decreases with increasing DL$x$ concentration (Supplementary Table 7) due to the aggregation of defects. After 12 runs of D-glucose selective hydrogenation, the size of dominant Ru NPs is enlarged (Fig. 2g–l and Supplementary Figs. 23 and 24), and the higher concentration of defects leads to the larger aggregation degree of Ru NPs in these two types of Ru-impregnated DEMOFs. However, the aggregation degree of Ru NPs in both Ru@$D_{1a-b}$ and Ru@$D_{2a-b}$ is lower than that in Ru@$D_0$ (Fig. 2g–i and Supplementary Table 6),

demonstrating both types of defects of low concentration can stabilize Ru NPs. Furthermore, the aggregation degree of Ru NPs in Ru@$D_{1a-c}$ is lower than that in Ru@$D_{2a-c}$ with the same $z$ of DL$x$, illuminating that the type-A defects can stabilize Ru NPs more efficiently than type-B defects against aggregation during the catalytic reaction, mainly due to the stronger anchoring effect between confined Ru NPs and basic pyridyl-N atoms at type-A defects. Consequently, the size, content, dispersion, and stability of Ru NPs, playing crucial roles in catalytic activity, can be controllably tuned by introducing different types of defects with varied DL$x$ concentrations.

**Probing two types of defects by UHV-FTIRS.** To gain detailed insight into the special modification effect of each type of implanted defects on the steric and electronic properties of Cr-CUSs as single active sites and Ru NPs in all Ru@DEMOFs, we employed ultra-high-vacuum Fourier transform infrared spectroscopy (UHV-FTIRS) using CO as a probe molecule, which is well suited to characterize the surface structural evolution and active sites of MOFs and oxide-based catalysts[38–43]. The UHV-FTIRS data of CO on Ru@$D_0$ at ~110 K show a main peak at 2201 cm$^{-1}$ with a shoulder at 2194 cm$^{-1}$ stemming from CO bound to pristine $Cr^{3+}$ MSAS (Fig. 3a). In addition, a weak band appears at 2153 cm$^{-1}$ being characteristic of CO adsorbed on the intrinsic defective $Cr^{\delta+}$-CUSs ($\delta < 3$) (Fig. 3b), in situ generated during synthesis[44,45]. For Ru@$D_{1a-c}$ (Fig. 3a, b), the $Cr^{3+}$-related CO vibrations remain unchanged in frequency (2199 cm$^{-1}$) while the CO bands originating from reduced $Cr^{\delta+}$ defects show a slight redshift with increasing doping level of DL1. In the case of Ru@$D_{2a-c}$ (Fig. 3c, d), both Cr-related CO bands shift slightly to a lower frequency with increasing $z$ of DL2. These results demonstrate the formation of electron-enriched $Cr^{\delta+}$ defects via the partial reduction of pristine $Cr^{3+}$-CUSs along with the incorporation of DL1 and DL2, which have different impacts on modifying MSAS as described in Fig. 1.

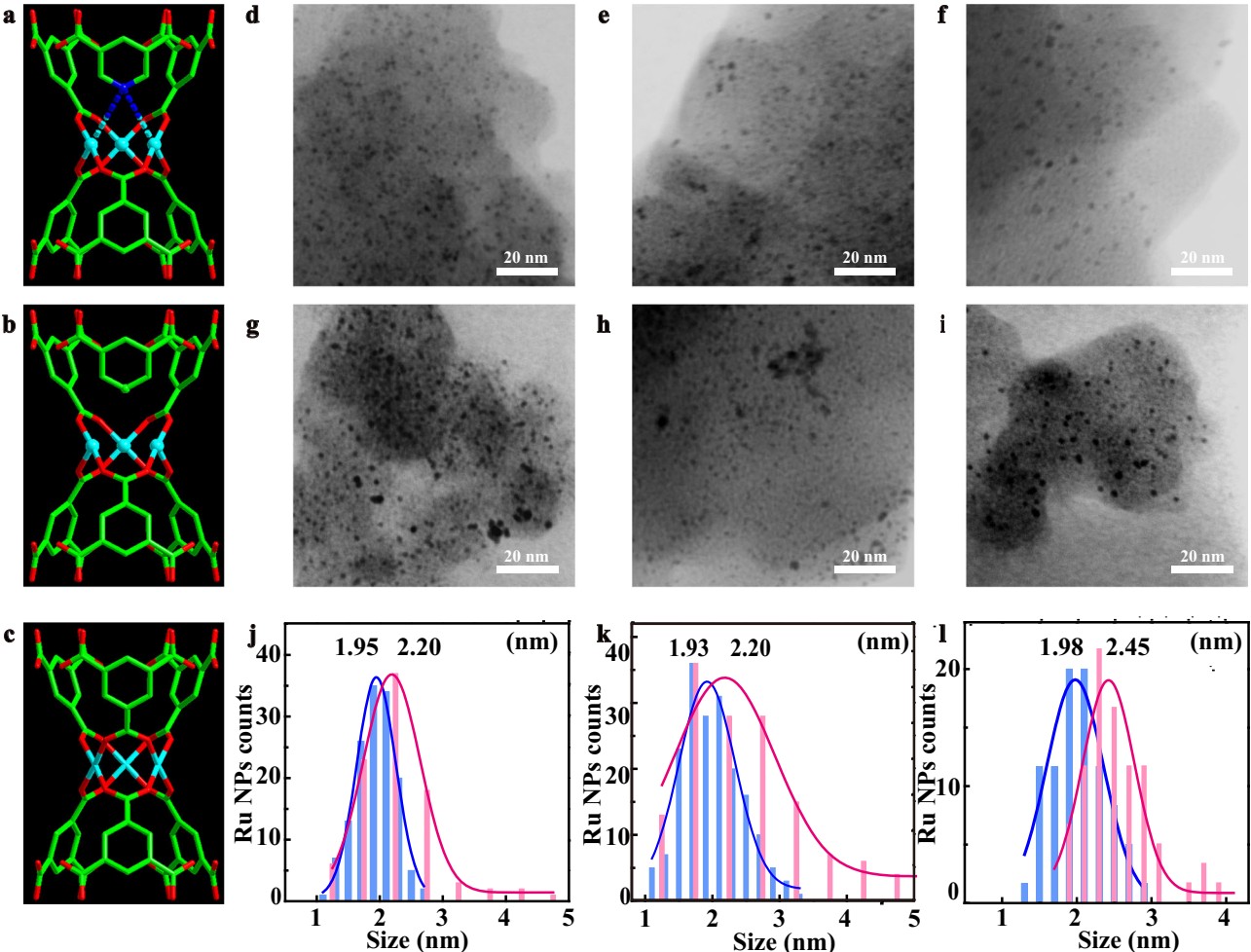

**Fig. 2 Identification of Ru NPs in Ru-impregnated parent and defect-engineered MIL-100-Cr MOFs. a–c** Schematic illustration of various Cr single active sites: MSAS with type-A defect in $D_{1a-c}$ and Ru@$D_{1a-c}$, formed via incorporation of DL1 with a weak ligator N at defect sites (**a**); MSAS with type-B defect in $D_{2a-c}$ and Ru@$D_{2a-c}$, generated by incorporation of DL2 without ligator at defect sites (**b**); pristine Cr single active sites (**c**). **d–f** STEM images of Ru@$D_{1a}$ (**d**), Ru@$D_{2a}$ (**e**), and Ru@$D_0$ (**f**) before catalyzing biomass hydrogenation of D-glucose to sorbitol. **g–i** STEM images of Ru@$D_{1a}$ (**g**), Ru@$D_{2a}$ (**h**), and Ru@$D_0$ (**i**), after catalyzing biomass hydrogenation of glucose to sorbitol for 12 runs. **j–l** Size distributions of Ru@$D_{1a}$ (**j**), Ru@$D_{2a}$ (**k**), and Ru@$D_0$ (**l**) before (blue) and after (pink) 12 runs of catalysis.

The controlled generation of defects via the above defect-engineering approach is further confirmed by a quantitative analysis of the IR data (Fig. 3e) that reveals a gradual increase in the intensity ratio of $Cr^{\delta+}$-CO to $Cr^{3+}$-CO species in $D_{2a-c}$ (Supplementary Fig. 27) as a function of the incorporated DL2 concentration. Importantly, the populations of $Cr^{\delta+}$-CUSs in both Ru@$D_{1a-c}$ and Ru@$D_{2a-c}$ are higher than that in $D_{2a-c}$, indicating that the Ru impregnation facilitates the formation of reduced $Cr^{\delta+}$ defects. This is primarily attributed to the fact that HCl molecules, produced by hydrolysis of $RuCl_3$ precursor, expand and enlarge the small-scale defects, similar to the water effect on $Zn_2(BDC-TM)_2(DABCO)$[3]. The lowered concentration of $Cr^{\delta+}$-CO in Ru@$D_{1b-c}$ and Ru@$D_{2b-c}$, in contrast with Ru@$D_{1a}$ and Ru@$D_{2a}$, is probably due to the lower stability of $D_{1b-c}$ and $D_{2b-c}$, as evidenced by the higher leaching amount of DLs and the larger decrease of BET surface areas after Ru NPs encapsulation (Supplementary Tables 2–4). These results demonstrate that the doping of Ru NPs inside DEMOFs is accompanied by the generation of additional defects.

We focus now on the spectral evolution at lower frequencies in the Ru-related CO vibration region with increasing the doping level of DL1 (Ru@$D_{1a-c}$, Fig. 3b) and DL2 (Ru@$D_{2a-c}$, Fig. 3d). The IR spectra are dominated by a rather broad feature ranging from 1980 to 2080 $cm^{-1}$, which varies in shape and intensity depending on the type and content of the incorporated DL. These signals are characteristic of CO species bonded linearly to Ru atop sites[46–48]. Fine structures are resolved only for Ru@$D_{1c}$ with higher DL1 concentration (Fig. 3b). Overall, the UHV-FTIRS data provide spectroscopic evidence for the presence of well-dispersed small Ru particles that feature a number of Ru sites with distinct steric (coordination) and electronic properties, thus leading to the overlapping of various CO bands bound to different Ru sites[46–48].

The temperature-dependent IR spectra allow us to gain more detailed insight into the active sites exposed by representative Ru@DEMOFs (Ru@$D_0$, Fig. 3f; Ru@$D_{1a}$, Fig. 3g; Ru@$D_{2a}$, Fig. 3h). The Cr-related CO bands disappear at about 240 K (Supplementary Fig. 28), revealing a weak interaction of CO with $Cr^{3+}$ and $Cr^{\delta+}$ MSAS with relatively low binding energies. In comparison, CO is more strongly bound to the confined Ru particles due to the substantially enhanced electron π back-donation. Close inspections of the IR data show a higher thermal stability of CO species adsorbed on Ru@$D_{1a}$ compared to Ru@$D_0$ and Ru@$D_{2a}$ where the CO signals vanish nearly completely at about 450 K. This finding reveals that the interaction between CO and Ru NPs is enhanced by the type-A defects featuring basic

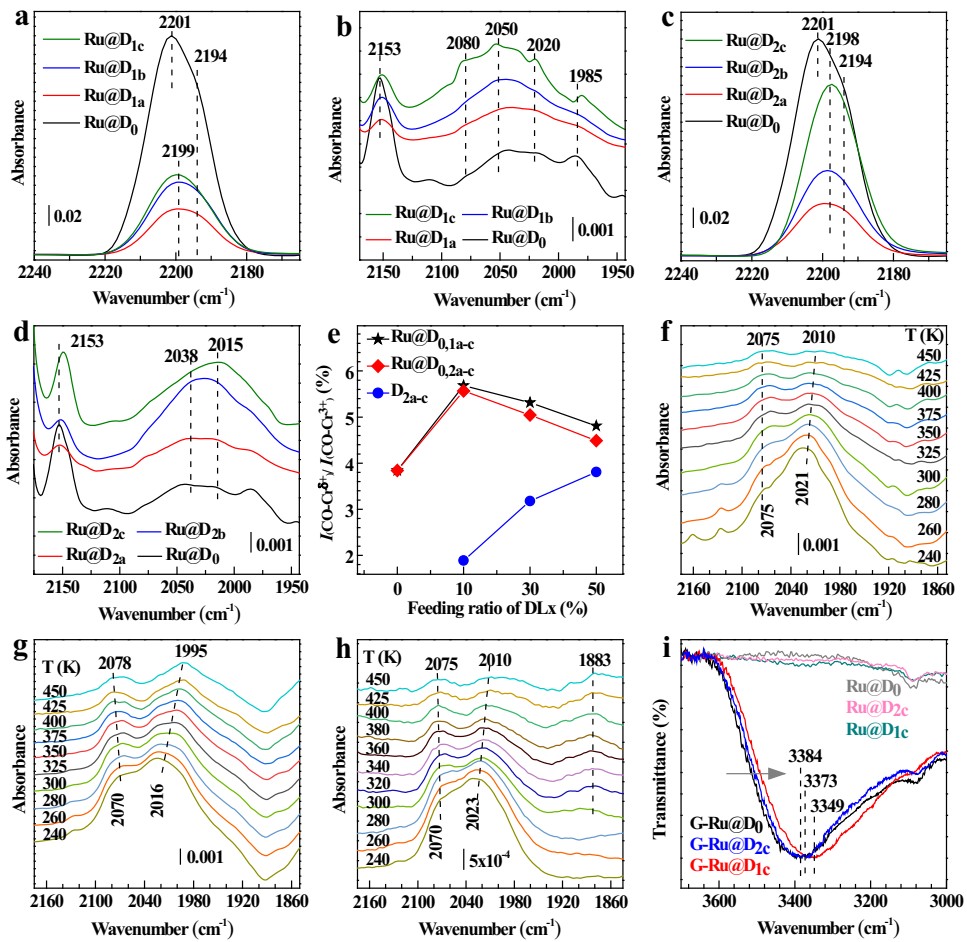

**Fig. 3 Experimental confirmation of the modification of Cr-MSAS and confined Ru NPs in Ru@DEMOFs by varying DLs type and concentration.**
**a–d** UHV-FTIR spectra obtained after CO adsorption (0.01 mbar) at ~110 K on Ru@D$_{1a-c}$ (**a**, **b**) and Ru@D$_{2a-c}$ (**c**, **d**) in different CO vibration regions.
**e** Integrated intensity ratio of Cr$^{\delta+}$ ($\delta < 3$)-related to Cr$^{3+}$-related CO bands as a function of the doping level of DLx. For comparison, the UHV-FTIRS data of CO on Ru@D$_0$ is also displayed in **a–d**. **f–h** Temperature-dependent UHV-FTIRS data of CO on Ru@D$_0$ (**f**), Ru@D$_{1a}$ (**g**), and Ru@D$_{2a}$ (**h**). Prior to exposure, each sample was heated to 500 K to remove all adsorbed species. **i** Routine FTIR spectra of the D-glucose impregnated G-Ru@D$_0$, G-Ru@D$_{1c}$, and G-Ru@D$_{2c}$, in comparison with the activated unloaded Ru@D$_0$, Ru@D$_{1c}$, and Ru@D$_{2c}$ samples.

pyridyl-N atoms. Despite the complex spectral pattern, two Ru-CO bands are resolved at 2016–2023 cm$^{-1}$ and 2070–2075 cm$^{-1}$ after heating the Ru@DEMOFs samples to 240 K (Fig. 3f–h). The occurrence of fine structures could be related to the thermal diffusion of CO to more stable sites, thus leading to an increase in ordering. The splitting of these two bands becomes slightly larger due to the modification of chemical environments along with the desorption of various CO species. The vibrational frequency and intensity ratio change depending on the type of DLx. The 2021 cm$^{-1}$ band is the predominant one for Ru@D$_0$ at 240 K, while the 2075 cm$^{-1}$ band becomes an intense one for Ru@D$_{1a}$ and Ru@D$_{2a}$. This higher frequency vibration is characteristic of the more positively charged Ru sites[46–50], which could be formed via the electronic interaction between embedded Ru NPs and Cr$^{\delta+}$-CUSs (Lewis acid sites) that lose one coordinating carboxylate. Furthermore, Ru@D$_{1a}$ shows a lower population of the 2075 cm$^{-1}$ band compared with Ru@D$_{2a}$ at higher temperatures, resulting from additional interaction between Ru particles and basic pyridyl-N atoms of DL1 that are absent in Ru@D$_{2a}$. Interestingly, a bridge Ru-related CO band is detected at 1883 cm$^{-1}$ as a weak signal for Ru@D$_{2a}$ (Fig. 3h), revealing the presence of larger Ru NPs as minor species in Ru@D$_{2a}$, in line with the STEM observation (Fig. 2k). These IR results validate that the special type of artificially introduced defects has a unique

regulation effect on the morphology, electronic, and steric properties of the embedded Ru particles. This is further supported by XPS investigations of Au@DEMOFs as a reference system (Supplementary Fig. 29).

To validate the effect of special types of implanted defects on the adsorption mode of D-glucose reactant, we have synthesized a series of D-glucose impregnated DEMOFs. The corresponding FTIR results (Fig. 3i and Supplementary Fig. 30) show that the broad band centered at ~3384 cm$^{-1}$, assigned to the OH stretching mode of D-glucose molecules in G-D$_0$ and G-Ru@D$_0$, shifts to a lower frequency after doping with DL1 and DL2. This is attributed to the weak interaction between D-glucose and reduced Cr$^{\delta+}$ MSAS. The OH vibrations of G-D$_{1c}$ and G-Ru@D$_{1c}$ show a further redshift in frequency compared to those of G-D$_{2c}$ and G-Ru@D$_{2c}$, revealing the formation of hydrogen bonds between hydroxyl groups in glucose molecules and basic pyridyl-N atoms of DL1 in Ru@D$_{1c}$.

**Influence of two types of defects on catalytic activity.** The comprehensive results of UHV-FTIRS, STEM, CO chemisorption, and ICP-OES demonstrate that the amount, dispersion, size, shape, and stability of Ru NPs as well as the binding strength between active sites and guest molecules, controlling adsorption and desorption rates, can be finely adjusted by rationally

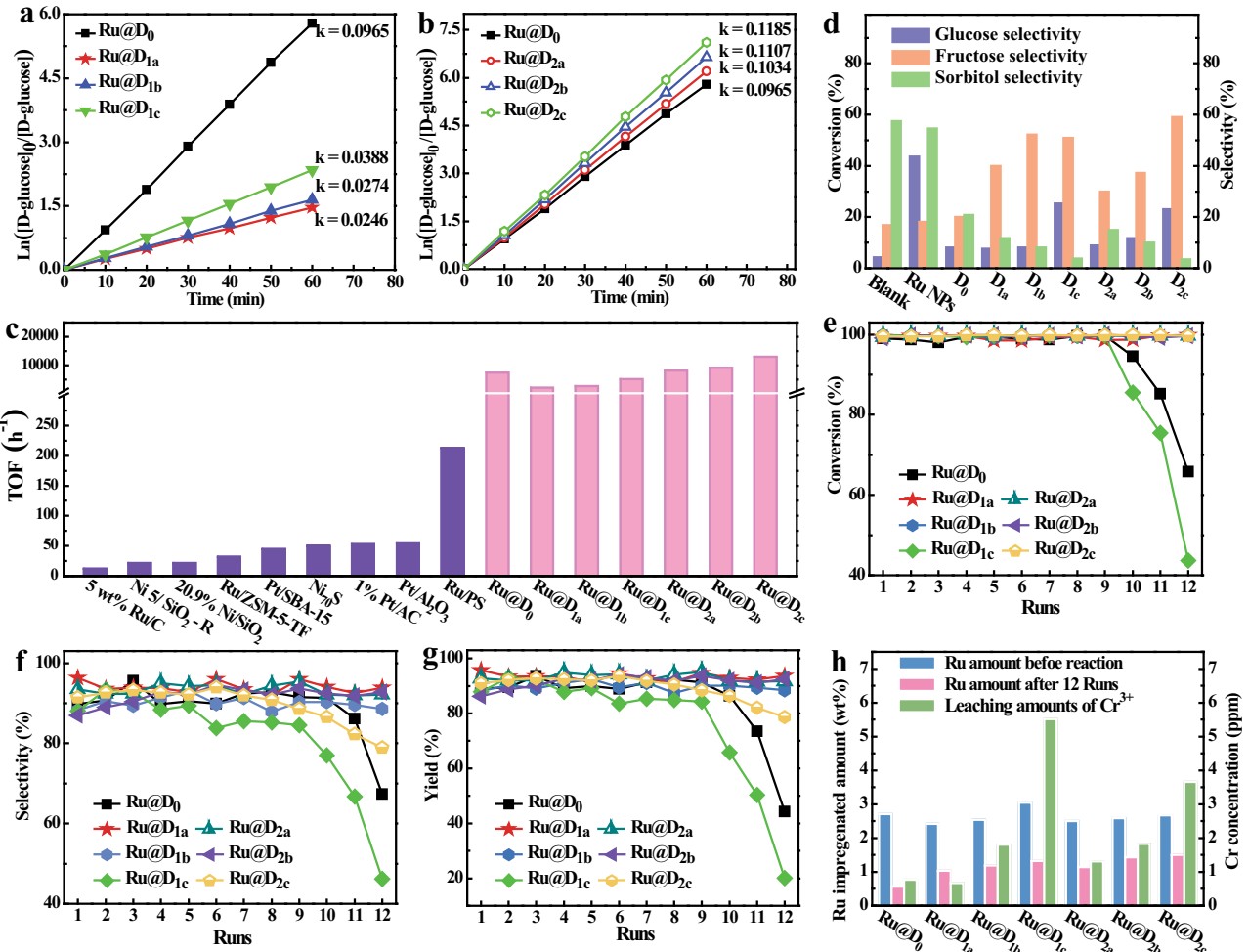

**Fig. 4 Catalytic activity and recyclability of Ru@D0 and Ru@DEMOFs catalysts. a, b** Kinetic studies of D-glucose selective hydrogenation reactions catalyzed by Ru@$D_0$ and Ru@$D_{1a-c}$ (**a**), by Ru@$D_0$ and Ru@$D_{2a-c}$ (**b**). **c** Comparison of TOF values between the Ru-impregnated MOFs catalysts in this work and various reported catalysts for the first run reaction. **d** Comparison of D-glucose conversion and selectivity to fructose and sorbitol without catalyst and catalyzed by pristine and defect-engineered MIL-100-Cr MOFs. **e–g** Comparison of D-glucose conversion (**e**), selectivity to sorbitol (**f**), and sorbitol yields (**g**) for the reactions catalyzed by Ru@$D_0$ and Ru@$D_{1a-c}$ and Ru@$D_{2a-c}$ for 12 recycles. **h** The impregnated amount of Ru NPs in Ru@$D_0$, Ru@$D_{1a-c}$, and Ru@$D_{2a-c}$ before and after catalyzing for 12 runs and the leaching amount of $Cr^{3+}$ ions after catalyzing for the third run. Reaction conditions: glucose aqueous solution (25 wt%, 1.543 mol/L, 50 g), catalyst amount (1 g), $H_2$ (5.0 MPa), temperature (120 °C), stirring rate (600 rpm), and reaction time (150 min for all Ru-impregnated MIL-100-Cr catalysts except Ru@$D_{1b}$ catalyzed reaction for 120 min).

implanting defects. The D-glucose hydrogenation follows the Langmuir–Hinshelwood mechanism, where the surface reaction is rate-determining step[51]. Therefore, it is expected that this reaction is primarily controlled by pore structures and binding strengths of MOFs catalysts with guest molecules. Moreover, the selectivity of sorbitol, limited by competitive side reactions including further isomerization of sorbitol, dimerization of D-glucose to fructose, and D-glucose hydrogenation to mannitol, could be also adjusted by the adsorption and desorption rate of reactants, intermediates, and products on MSAS. Consequently, the activity, selectivity, and recyclability of Ru-impregnated MIL-100-Cr is anticipated to be improved through artificially implanted defects.

To prove the above assumption, the catalytic performance of Ru@$D_0$, Ru@$D_{1a-c}$, and Ru@$D_{2a-c}$ toward D-glucose hydrogenation have been comprehensively investigated. The optimized reaction conditions were discussed in detail in Supplementary Section 2, and the following comparisons of the D-glucose conversion, sorbitol selectivity, and yield of all catalysts were performed under respective optimal reaction conditions.

All apparent D-glucose hydrogenation reaction orders are determined to be pseudo-first order (Fig. 4a, b). The largest BET-specific surface areas and the most hierarchical mesopores of Ru@$D_{1a}$ should result in the fastest reaction rate of Ru@$D_{1a}$ among Ru@$D_0$ and Ru@$D_{1a-c}$. However, the rate constant, defined as $k$, of Ru@$D_{1a}$ is much lower than that of Ru@$D_0$ (0.0965) (Fig. 4a). This could be explained by the stronger binding strength of Ru NPs and guest molecules that leads to partial blocking of available active sites. The $k$ values of Ru@$D_{1a-c}$ (0.0246–0.0388) are all lower than that of Ru@$D_0$, mainly due to the reduced electron density of Ru NPs and the presence of hydrogen bonding between reactant molecules and basic pyridyl-N atoms at defect sites of Ru@$D_{1a-c}$, as described in Fig. 1f. In contrast, the $k$ values of Ru@$D_{2a-c}$ (from 0.1034 to 0.1185) are substantially higher than that of Ru@$D_0$ (Fig. 4b), which is primarily attributed to the weaker binding of guest molecules and $Cr^{\delta+}$ MSAS and the absence of interactions between the defect sites of DL2 and adsorbed molecules, thus accelerating the desorption of reaction products, as proposed in Fig. 1g. The stepwise increased reaction rates of both Ru@$D_{1a-c}$ and Ru@$D_{2a-c}$

are mainly due to the gradual increase of the loading amount of Ru NPs upon increasing the content of DL$x$, which facilitates the activation of dihydrogen to H atoms. Moreover, the evolution of turnover frequency (TOF) numbers is basically consistent with the evolution of reaction rate for these Ru@DEMOFs (Fig. 4c), illustrating that the reactivity of Ru@DEMOFs can be regulated in a controllable manner by artificially implanting defects that vary in type and concentration. Noticeably, even the TOF value (2150.73 h$^{-1}$) of Ru@D$_{1a}$, being the smallest one among all these Ru-impregnated MOF catalysts, is significantly larger than the value of all kinds of previously reported catalysts (TOF = 235 h$^{-1}$, Fig. 4c and Supplementary Table 9)[1], revealing that these Ru-impregnated MIL-100-Cr catalysts exhibit excellent activity toward D-glucose selective hydrogenation.

**Determination of the roles of each active species and their synergistic catalytic mechanism.** To gain an in-depth understanding of the reaction mechanism, we have further investigated the role of pristine MSAS, artificially implanted defects, and well-dispersed Ru NPs as well as their synergetic effect on the high conversion (>98%) of D-glucose and selectivity to sorbitol (>85%). Without any catalyst, the conversion of D-glucose is as low as 4.22%, accompanied by a 57.45% selectivity to sorbitol and 16.91% selectivity to mannitol (Fig. 4d). When D$_0$ containing a little amount of intrinsic defects (Fig. 3d) is used as a catalyst, the conversion of D-glucose increases to two times that without a catalyst, accompanied by a slightly improved selectivity of fructose but substantially reduced selectivity toward sorbitol by ~2 times (Fig. 4d), illuminating that the MIL-100-Cr MOF can promote the D-glucose conversion and facilitate the formation of fructose, but restrain the sorbitol production.

For all DEMOFs, the selectivity to fructose is much higher while selectivity to sorbitol is lower compared with the pristine D$_0$. Upon increasing the incorporated amount of DL$x$, the conversion of D-glucose and the selectivity to fructose show an overall stepwise increase, while the selectivity of sorbitol gradually decreases. When $z$ reaches 50%, the D-glucose conversion is boosted about 2 times, and the selectivity to fructose increases by ~1.5 and 2 times for D$_{1c}$ and D$_{2c}$, respectively. In contrast, the selectivity to sorbitol is reduced to 1/5 for both of them (Fig. 4d) in relation to that of the reaction over D$_0$. Evidently, both type-A and type-B defects can boost the activity of MIL-100-Cr to the D-glucose hydrogenation reaction and the selectivity to fructose, but reduce the selectivity to sorbitol. This is probably attributed to the weak coordination bonds between D-glucose and defective MSAS. Noticeably, type-A and type-B defects show the distinguishing disparity in the improvement of catalytic performances, which is mainly attributed to the hydrogen bonds between hydroxyl groups in glucose molecules and basic pyridyl-N atoms in DL1 incorporated into the framework of Ru@D$_{1a-c}$. Compared with the blank reaction in absence of any catalysts, Ru NPs greatly increase the D-glucose conversion by more than 10 times (Fig. 4d), but show a negligible effect on the selectivity of sorbitol and fructose. These results identified that the confined Ru NPs boost the reaction activity as they can decompose H$_2$ into active H atoms. In comparison with that of Ru NPs, parent MOF and DEMOF catalysts, the higher D-glucose conversion and sorbitol selectivity of Ru@D$_0$, Ru@D$_{1a-c}$, and Ru@D$_{2a-c}$ can be attributed to the synergistic catalysis of the embedded small Ru particles, defect sites and pristine MSAS.

The above comprehensive results provide solid evidence for the following synergistic reaction mechanism of D-glucose selective hydrogenation to sorbitol catalyzed by Ru@DEMOFs (Fig. 1): (1) H$_2$ molecules are first activated on the surface of Ru NPs and then dissociated into active H atoms to participate in the subsequent D-glucose hydrogenation reaction. (2) D-glucose molecules absorb on the surface of D$_{1a-c}$ with type-A defects via coordination bonds between D-glucose and Cr-MSAS as well as hydrogen bonds between D-glucose and basic pyridyl-N atoms at defect sites, having decisive importance in controlling the speed of adsorption and desorption of reactants, intermediates and products on Ru@DEMOFs. (3) The activated H atoms transfer to D-glucose molecules with subsequent hydrogenation to form sorbitol. (4) Finally, the target product sorbitol desorbs from the catalyst surface (Fig. 1f, g), and the Ru@DEMOFs catalysts are recovered to the original state for processing the next catalytic cycle. We note that the catalytic mechanism of Ru@D$_{2a-c}$ is similar to that of Ru@D$_{1a-c}$, and the significant difference between them is the absence of hydrogen bonds between D-glucose and the defect sites of DL2 in Ru@D$_{2a-c}$ (Fig. 1g).

**Effect of two types of defects on recyclability.** To investigate the special effect of each type of defects on the recyclability of Ru-impregnated MIL-100-Cr MOF catalysts, the D-glucose selective hydrogenation reactions were tested for 12 runs. In the first run, Ru@D$_{1a}$ exhibits the highest sorbitol yield (~96%) among all Ru-impregnated MOF catalysts tested herein, while Ru@D$_{2a}$ shows a higher sorbitol yield than Ru@D$_{2b-c}$ and Ru@D$_0$, probably due to their higher concentration of Cr$^{\delta+}$-MSAS and BET surface areas. The higher sorbitol yield of Ru@D$_{1a}$ than that of Ru@D$_{2a}$ illuminates that type-A defects are more efficient at improving the reactivity of Ru-impregnated MIL-100-Cr via the synergistic effect of defects, MSAS, and Ru NPs. After 12 runs, the selectivity to sorbitol and D-glucose conversions of Ru@D$_{1a-b}$ and Ru@D$_{2a-b}$ maintain unchanged. Only Ru@D$_{1c}$ and Ru@D$_0$ show a decrease in D-glucose conversion (Fig. 4e), while for Ru@D$_{2c}$, Ru@D$_0$, and Ru@D$_{1c}$, there is a decrease in selectivity and yield to sorbitol after recycling 12 times (Fig. 4f, g). These results demonstrate that the incorporation of DL1 and DL2 in the framework of MIL-100-Cr with low $z$ (≤30%) can enhance the recyclability of Ru@DEMOFs catalysts.

To further figure out the reasons for the disparity of recyclability of these catalysts, the leaching amounts of Ru and Cr elements after catalysis have been investigated. As shown in Fig. 4h, the leaching amount of Cr ions of Ru@D$_{1b-c}$ and Ru@D$_{2b-c}$ after three cycles of catalysis gradually increase upon increasing $z$ of both DL1 and DL2, illuminating a gradual decrease of the framework stability upon increasing the doping level of DL$x$. Noticeably, the leaching amount of Cr ions in Ru@D$_0$ are significantly higher than that of Ru@D$_{1a}$, but much lower than that of Ru@D$_{1b-c}$ catalysts. These results reveal that the robustness of the catalyst framework has been enhanced at $z$ of DL1 ≤10%, but declined upon further increase $z$ of DL1. In sharp contrast to DL1, the incorporation of DL2 exhibits a negative effect on the skeleton stability of Ru@D$_{2a-c}$. These findings together with the results of PXRD (Supplementary Fig. 1) validate the weak coordination bonds between the pyridyl-N atoms (Lewis base) in DL1 and Cr ions (Lewis acid, Fig. 1b) resulting in higher stability of the skeleton of D$_{1a-c}$ compared with D$_{2a-c}$. For all the Ru@DEMOFs catalysts after 12 runs of reaction, the leaching amounts of Ru are in the range of 43.47–61.97% (Supplementary Table 5), being far less than that of Ru@D$_0$ (78.97%), while the maintained amount of Ru for each type of Ru@DEMOFs is enhanced with increasing $z$ of DL$x$, confirming the anchoring effect of defect sites on Ru NPs to prohibit the leaching of Ru from the MOFs during catalysis. Consequently, the low recyclability of Ru@D$_0$ can be attributed to its lowest residual content of embedded Ru NPs, while in the case of Ru@D$_{1c}$ and Ru@D$_{2c}$, the low recyclability should be related to their framework fragility due to large DL$x$ incorporation. Moreover, the high

aggregation degree of Ru NPs in Ru@$D_0$, Ru@$D_{1c}$, and Ru@$D_{2c}$ after catalysis for 12 runs are also assigned to their low recyclability. After 12 runs of catalysis, Ru@$D_{1a}$ exhibits the highest sorbitol yield (~96%) and the best recyclability due to the lowest aggregation of Ru NPs and the highest framework robustness of Ru@$D_{1a}$ among all Ru-impregnated MIL-100-Cr catalysts tested herein. These results illuminate that the catalytic performance of the Ru-impregnated MIL-100-Cr catalysts can be improved by rational design and control of defects.

In summary, we designed and controllably implanted two types of defects in the cationic framework of MIL-100-Cr to impregnate Ru NPs for simultaneously boosting the selectivity, reactivity, and reusability of the resulting catalysts toward D-glucose selective hydrogenation to sorbitol. The synergistic catalysis mechanism of Ru@DEMOFs was clarified based on in-depth investigations of the nature of confined small Ru NPs and defect sites, as well as the findings that Ru NPs, MSAS, and artificially implanted defects can increase the D-glucose conversion while MSAS and defects can boost the selectivity of fructose but degrade that of sorbitol. This work provides an avenue for designing and evolving MOF-based catalysts with excellent catalytic performance for target biomass conversion reactions.

## Methods

**Synthesis of MIL-100-Cr pristine MOF and DEMOFs**. In all, 6.24 mmol $CrO_3$ (624 mg) and 6.25-n mmol $H_3$btc, n mmol $H_2DLx$ ($x = 1$: 3, 5-pyridinedicarboxylic acid; $x = 2$: 1, 3-benzenedicarboxylic acid, $n = 0$, 0.625, 1.875 and 3.125 mmol, respectively) were dissolved in water (30 mL), followed by adding 8.28 mmol hydrofluoric acid aqueous solution (40%, 0.36 mL) with stirring. The mixture was stirred for 20 min, and then was transferred to a Teflon-lined stainless-steel reactor, which was heated at 220 °C for 96 h, subsequently, was cooled to room temperature. All the collected microcrystalline powders via centrifugation were immersed in ethanol at 80 °C for 24 h, and then cooled to 50 °C. The obtained samples were recollected via centrifugation, followed by being washed with ethanol three times to remove all residual precursor species. After dried in an oven at 80 °C under air for 12 h, all obtained samples were activated at 180 °C under vacuum (~$10^{-3}$ mbar) for 24 h. Finally, all prepared samples were stored under nitrogen in a glove box for further tests.

**Synthesis of Ru-impregnated MIL-100-Cr pristine MOF (Ru@$D_0$) and DEMOFs catalysts (Ru@$D_{1a-c}$ and Ru@$D_{2a-c}$)**. In all, 30 mL ethanol solution containing $x$ mmol $RuCl_3$ ($x = 0.099$, 0.148, 0.198, 0.247, and 0.495, respectively) was added drop wisely into 1.0 g MOF samples under vigorous stirring for 30 min, and then the obtained mixture was stirred for 16 h under reflux. After alternatively washing with ethanol and deionized water three times, the resulting catalyst precursors were then reduced under 4 MPa hydrogen pressure at 120 °C for 180 min in a stainless autoclave to obtain a series of Ru-impregnated MOF catalysts containing different mass percentage of Ru NPs.

**Synthesis of Au impregnated MIL-100-Cr pristine MOF (Au@$D_0$) and DEMOFs catalysts (Au@$D_{1c}$ and Au@$D_{2c}$)**. In all, 4.314 mL ethanol solution containing sodium chloraurate (0.0214 mol/L) was added into 300 mg dry and activated MIL-100-Cr MOF under $N_2$ sealed in a 10 mL glass tube by syringe. The resulting mixture was heated at 75 °C for 1 h to ensure well distribution of sodium chloraurate inside the framework. The resulting sample was collected by centrifuged at 13,528 ×g for 10 min, subsequently was washed with ethanol and dried at 80 °C for 4 h, and then activated at 220 °C for 12 h. Finally, the obtained dry samples were reduced under 5 MPa hydrogen pressure at 120 °C for 360 min in a stainless autoclave to obtain Au (5 wt%) impregnated MOF materials.

**Synthesis of glucose impregnated Ru@MIL-100-Cr pristine MOF (G-Ru@$D_0$) and Ru@DEMOFs catalysts (G-Ru@$D_{1c}$ and G-Ru@$D_{2c}$)**. In all, 4 mL glucose aqueous solution (5.55 mol/L) was added into 50 mg dry and activated Ru@MOFs under $N_2$ sealed in a 5 mL glass tube by syringe, and followed by ultrasound for 10 min. The mixture was then transferred into a three-necked flask (15 mL) under heating at 100 °C for 15 min with vigorous mechanical stirring to ensure the well distribution of glucose molecules inside the framework. The resulting sample was collected by centrifuging at 13,527 ×g for 10 min, and then washed with water to remove glucose covered on the surface of the sample, finally dried at 80 °C for 12 h. Before testing routine FTIR in $N_2$ protection glove box, the glucose impregnated Ru@$D_0$, Ru@$D_{1c}$, and Ru@$D_{2c}$ samples were activated at 160 °C for 12 h.

## Data availability

All data necessary to support the conclusions can be found in the paper or its Supplementary Information. All data required to verify and interpret the findings are available from the corresponding authors upon request.

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

## Acknowledgements

This work was financially supported by National Natural Science Foundation of China (21871139, receipted by Z.F. and 21975122, receipted by Q.J.), Natural Science Foundation of Jiangsu Province, China (BK20191360, receipted by Q.J.). J.J.W. and Y.M.W. acknowledge the financial support by the German Research Foundation (DFG) through projects 392178740 and 426888090 (SFB 1441).

## Author contributions

Z.F. proposed the concept of this manuscript. Z.F. and Y.W. supervised this work and were responsible for analyzing the data and composing the manuscript. W.X. and Y.Z executed the syntheses and evaluated catalytic performance, meanwhile, collected PXRD, FTIR, and chemisorption data. Y.Z was also responsible for STEM, EA, NMR, and ICP-OES data. Y.X. collected BET data, and L.B. designed the concept diagrams. J.W. performed the in situ UHV-FTIR and XPS experiments and analyzed the data. Q.J. gave valuable comments and checked this paper. All authors were involved in the composing of the manuscript and have given consent to this publication.

## Competing interests

The authors declare no competing interests.
