## [Peer Review File · Nature Communications]

Defects engineering simultaneously enhances activity and recyclability of MOFs in selective hydrogenation of biomassREVIEWER COMMENTS

Reviewer #1 (Remarks to the Author):

1. Please explain the mechanism about the introduced defects constructed by the incorporation of DLx with low concentration can enhance the framework stability of MIL-100-Cr. What is the optimal concentration of DLx during 1-10%? The data should be shown in Supplementary Materials.
2. Fig. 3e and f should be redrawn. Some marked peaks in Fig. 3e and f are different with the discussion in manuscript which makes the reading difficult;
3. There is no any discussion about Supplementary Fig. 16;
4. There is an error in Fig. 4e. It should be leaching amounts of Cr³⁺ ions rather than Cr residue.
5. Fig. 4h should be redrawn. The data of Ru@D1a can not be found in figure.
6. The order of figures are chaotic, especially Fig. 3 and Fig. 4. Rearrangement of all figures is necessary. The figure should be in the order as the discussion in manuscript.
7. Please explain the synergistic catalytic mechanism between Ru NPs, MSAS, and DLx detailedly.
8. More reference about synergistic effect could be considered: for example, Adv Mater., 2019, 31,52,1904969 and Nature communications, 2021, 12(1): 1-9.

Reviewer #2 (Remarks to the Author):

The authors present a study of how defect engineering a MOF can have an influence on the incorporation of Ru nanoparticles and the consequent catalytic activity towards a hydrogenation reaction.

A lot of high-quality work has been done here and there are some interesting conclusions about the influence on artificially-induced defects in MOFs. Whilst this has been reported by other researchers, I believe this is a noteworthy contribution to the body of literature.

I have a few issues with some of the interpretation of the data, specifically:

1) p4 l17 "The high quality UHV-FTIRS at ~110 K of Ru@D0 show a main sharp peak at 2201 cm⁻¹ with a shoulder band peaked at 2194 cm⁻¹ stemming from COCr³⁺ (Fig. 3a, b) and a weak band centered at 2153 cm⁻¹ assigned to CO adsorbed on the intrinsic defective Cr^{δ+}-CUSs ($\delta < 3$) (Fig. 3e, f), in situ generated during synthesis."

This assignment is made without any justification. the authors need to refer to previously published literature showing similar absorbances in a related system

2) p5 l13 "These results validate our proposal that the special type of artificially introduced defects have a unique regulation effect on the modality of Ru NPs."

I don't understand what the authors mean by "regulation effect on the modality". Furthermore, the authors need to explain in the text above this sentence what the physical significance of the loss of spectral features in the FTIR spectra in figures 3 e and f is.

3) p6 line 5

"The higher stability of CO absorbed on the Ru NPs can be assigned to the reduced electron density of Ru NPs in Ru@D1a, confirmed by XPS measurements (Supplementary Fig. 18)"

The XPS in figure S18 DOES NOT support this conclusion. Their conclusion is based on the exact position of the Ru3d5 peak which is barely discernible over the much stronger C1s peak in the

spectra (particularly in subfigures a and b). Any measurement of peak position is therefore subject to massive uncertainty and consequently they can't draw any conclusions about charge transfer to/from the particles from this data. This analysis MUST be removed.

In summary, i believe this paper is suitable for publication in Nature Comms, subject to the above revisions being made.

Reviewer #3 (Remarks to the Author):

In this work, the authors constructed a series of defect Ru@DEMOFs composite materials via introducing different amounts of isophthalic acid or 3,5-pyridinedipyridine in the MIL-100(Cr). Moreover, compared to the parent catalyst Ru@D0, the defect Ru@DEMOFs has shown high catalytic performance and recoverability in the selective hydrogenation transformation of D-glucose. For further investigation, it is found that the synergistic effect of small defective molecules, Ru nanoparticles and metal nodes is the main factor to achieve the high catalytic performance. I think the authors have carefully characterized and discussed the defects. Unfortunately, from the perspective of the overall work, the current manuscript is not acceptable for Nature Communication. The reasons are listed as follow:

1. In the synthesis part, the mentioned method of generating defects via mixing the dicarboxylic acids in MOF has been widely reported. Therefore, the synthetic method in this work is lack of novelty. At the same time, the XRD results indicate that, after forming the defective MOF, the structure of MIL-100 has an apparent collapse, especially in the D1b-c and the D2b-c samples. I am afraid that the crystalline structures of defective MOFs with 30% or 50% doping dicarboxylic acid cannot be maintained. To make sure the MOF structure well remains, it is suggested to reduce the amount of doping ligand. Moreover, the dicarboxylic acid amount in the defective MOF is not determined? This should be fixed.
2. I found that the Ru nanoparticles are obtained via the impregnation method. This ship-in-a-bottle method results in the different size distributions of Ru NPs in each defective MOF, which has influence on the electronic state of Ru NPs and the final catalytic performance. Therefore, control experiments should be carried out to exclude particle size effects during the reaction.
3. The authors mentioned that the surface electronic state of Ru nanoparticles is changed in the defective MOFs, but how electron transfer affects the catalytic performance has not been verified. This is an important factor for the catalytic performance – it is necessary to demonstrate this point. In addition, it is assumed that the presence of H-bond in the defect MOF (D1a-c) has interaction with the substrate leading to the high catalytic performance. I think it is necessary to conduct the control experiments and provide related evidence to confirm this point. Moreover, a reliable reaction mechanism of D-glucose transformation in this work is expected.
4. In the recyclability test, please give the TEM results of Ru@D0 after 12 runs of reaction.

Dear reviewers,

Thank you very much for your positive feedback and constructive comments that have improved the clarity and raised the impact of our manuscript. We have very carefully studied your comments and subjected the manuscript to a major revision. We sincerely hope that we have adequately accounted for all your concerns by providing additional experimental data and intensive discussion. All changes in the main text and supplementary information are highlighted in blue, and the point-by-point responses to all your comments are listed below:

Reviewer #1 (Remarks to the Author):

1. Please explain the mechanism about the introduced defects constructed by the incorporation of DLx with low concentration can enhance the framework stability of MIL-100-Cr. What is the optimal concentration of DLx during 1-10%? The data should be shown in Supplementary Materials.

Answer: Defective linkers (DLs) can act as coligand to stabilize the framework of DEMOFs. Moreover, it has been reported that DLs show coordination modulation effect on the crystal growth of MIL-100-Cr DEMOFs at molecular level during in-situ synthesis, which also plays a critical role in the stability of the obtained MIL-100-Cr DEMOFs (*J. Am. Chem. Soc.* 2020, 142, 3174-3183, *Angew. Chem. Int. Ed.* 2017, 56, 563-567)^{1, 2}. Therefore, the stability of DEMOFs can be tuned by incorporation of defective ligands that vary in both the type and concentration. In this work, we found that the introduced defects constructed by the incorporation of DLx with low concentrations can enhance the framework stability of MIL-100-Cr¹. We included a sentence in the revised manuscript to make this point clear (line 16, column right, page 3).

In order to further find out the optimal feeding ratio of DLx to TL (z) ranging from 1% to 10%, we have conducted additional experiments and synthesized two types of DEMOFs and Ru@DEMOFs with z to be 5%. The corresponding results are shown in Supplementary Tab. 10 and Supplementary Fig. 42 (please see below). The new results reveal that for the Ru@D_{1a'} and Ru@D_{2a'} DEMOF catalysts, the D-glucose conversion is comparable to that acquired for Ru@D₀, Ru@D_{1a} and Ru@D_{2a} DEMOFs, while the selectivity to sorbitol and the sorbitol yield are higher than that of Ru@D₀, but lower compared

to Ru@D_{1a} and Ru@D_{2a}. The rate constant (*k*) and turnover frequency (TOF) of Ru@D_{1a'} and Ru@D_{2a'} are consistent with the expected values deduced from the evolution trends of *k* and TOF numbers upon the increase of incorporated defective ligands. A systematic comparison of D-glucose conversion (a), selectivity to sorbitol (b) and sorbitol yields (c) for D-glucose selective hydrogenation reaction catalyzed by these DEMOF samples for 12 recycles shows that the recyclability of Ru@D_{1a'} and Ru@D_{2a'} is higher than that of Ru@D₀ but lower compared with Ru@D_{1a} and Ru@D_{2a}. Overall, these results demonstrate that the optimal feeding molar ratio of DLs to total ligands for these two types of Ru-impregnated DEMOFs is 10%. The full set of data and detailed discussion on these new results have been added in Supplementary Information.

Supplementary Table 10 The list of D-glucose conversion, selectivity and yield of sorbitol, rate constant (*k*) and turnover frequency (TOF) for Ru@D₀, Ru@D_{1a'}, Ru@D_{1a}, Ru@D_{2a'} and Ru@D_{2a} toward the hydrogenation of D-glucose to sorbitol reaction for the first recycle.

	D-glucose conversion (%)	Selectivity (%)	Yield (%)	Rate constant (k)	Turnover frequency (TOF)
RuD ₀	99.071	89.826	88.991	0.0965	7415.2326
Ru@D _{1a'}	98.845	91.529	90.472	0.0237	1802.6575
Ru@D _{1a}	99.372	96.331	95.726	0.0246	2150.7299
Ru@D _{2a'}	99.283	92.318	91.657	0.0989	7495.4229
Ru@D _{2a}	99.343	93.417	92.803	0.1034	8201.4875

Supplementary Fig. 42 Comparison of D-glucose conversion (a), selectivity to sorbitol (b) and sorbitol yields (c) for hydrogenation of D-glucose to sorbitol reactions catalyzed by Ru@D₀, Ru@D_{1a'}, Ru@D_{1a},

Ru@D_{2a'} and Ru@D_{2a} for 12 recycles. Reaction conditions: glucose aqueous solution (25 wt%, 1.543 mol/L, 50 g), catalyst amount (1 g), H₂ (5.0 MPa), temperature (120 °C), stirring rate (600 rpm) and reaction time (150 minutes).

2. Fig. 3e and f should be redrawn. Some marked peaks in Fig. 3e and f are different with the discussion in manuscript which makes the reading difficult;

Answer: Thank you very much for your careful reading and corrections. We have carefully re-analyzed the UHV-FTIRS data and added in the revised Fig. 3 new results including temperature-dependent UHV-FTIRS data of CO on various Ru@DEMOFs as well as routine FTIR spectra of D-glucose impregnated Ru@DEMOFs and G-Ru@D_{2c}.

Main text Fig. 3 | Experimental confirmation of the modification of Cr-MSAS and confined Ru NPs in Ru@DEMOFs by varying DLs type and concentration. UHV-FTIR spectra obtained after CO adsorption (0.01 mbar) at ~ 110 K on **a, b** Ru@D_{1a-c} and **c, d** Ru@D_{2a-c} in different CO vibration regions. **e** Integrated intensity ratio of Cr ^{δ +} ($\delta < 3$)-related to Cr³⁺-related CO bands as a function of the doping level of DLx. For comparison, the UHV-FTIRS data of CO on Ru@D₀ is also displayed (**a-d**). **f-h** Temperature-dependent UHV-FTIRS data of CO on **f** Ru@D₀, **g** Ru@D_{1a}, and **h** Ru@D_{2a}. Prior to exposure, each sample was heated to 500 K to remove all adsorbed species. **i** Routine FTIR spectra of the D-glucose impregnated G-Ru@D₀, G-Ru@D_{1c} and G-Ru@D_{2c}, in comparison with the activated unloaded Ru@D₀, Ru@D_{1c} and Ru@D_{2c} samples.

3. There is no any discussion about Supplementary Fig. 16;

Answer: We have followed your valuable suggestions and added the discussion about Supplementary Fig. 16 in the revised Supplementary Information (now Supplementary Fig. 29, page

37).

4. There is an error in Fig. 4e. It should be leaching amounts of Cr^{3+} ions rather than Cr residue.

Answer: Thank you for your remark. The term “Cr residue” has been corrected to “Leaching amounts of Cr^{3+} ions” in the main text.

5. Fig. 4h should be redrawn. The data of $\text{Ru}@D_{1a}$ cannot be found in Fig. 4h.

Answer: The data of $\text{Ru}@D_{1a}$ has been added in the updated Fig. 4h according to your valuable suggestions.

6. The order of figures is chaotic, especially Fig. 3 and Fig. 4. Rearrangement of all figures is necessary. The figure should be in the order as the discussion in manuscript.

Answer: We have carefully checked and rearranged the order of figures in the manuscript.

7. Please explain the synergistic catalytic mechanism between Ru NPs, MSAS, and DLx detailed.

Answer: We have conducted additional experiments in order to gain detailed information about the unique effects of each type of implanted defects on the adsorption mode of D-glucose reactant and thus to further investigate the synergistic catalytic mechanism between Ru NPs, MSAS, and DLx towards the D-glucose hydrogenation reaction. Firstly, we have systematically synthesized a series of D-glucose impregnated DEMOF samples including D_0 (G- D_0), D_{1c} (G- D_{1c}), D_{2c} (G- D_{2c}), $\text{Ru}@D_0$ (G- $\text{Ru}@D_0$), $\text{Ru}@D_{1c}$ (G- $\text{Ru}@D_{1c}$) and $\text{Ru}@D_{2c}$ (G- $\text{Ru}@D_{2c}$). In the next step, these samples were characterized by FTIR spectroscopy, and the corresponding results are shown in Fig. 3i in the main text and Supplementary Fig. 32. Clearly, the broad band centered at $\sim 3384 \text{ cm}^{-1}$, assigned to the -OH stretching mode of D-glucose molecules in G- D_0 and G- $\text{Ru}@D_0$, shifts to lower frequency after doping with defective linkers (DL1, DL2), demonstrating the presence of weak coordination bonds between D-glucose and defective MSAS. Interestingly, the -OH vibrations of G- D_{1c} and G- $\text{Ru}@D_{1c}$ shows a further redshift in frequency compared to those of G- D_{2c} and G- $\text{Ru}@D_{2c}$, revealing the formation of hydrogen bonds between hydroxyl groups in glucose molecules and basic pyridyl-N atoms in DL1

incorporated into the framework of Ru@D_{1c}. Moreover, the disparity of OH-bands between the G-DEMOfs and G-Ru@DEMOfs further indicates that the Ru NPs impregnation process results in the evolution of both types of defects. Overall, these results validate our proposed synergistic catalytic mechanism of D-glucose selective hydrogenation to sorbitol for the two different kinds of Ru-NPs impregnated DEMOfs (Fig. 1 in Main text).

To get an in-depth understanding of the reaction mechanism, the role of pristine MSAS, artificially implanted defects and Ru NPs as well as their synergetic effect on the high conversion (> 98%) of D-glucose and selectivity to sorbitol (> 85%) during the catalytic process have been investigated. Without any catalyst, the conversion of D-glucose is as low as 4.22%, accompanied by a 57.45% selectivity to sorbitol and 16.91% selectivity to mannitol (Fig. 4d in Main text). When D₀ containing a little amount of intrinsic defects (Fig. 3d in Main text) is used as catalyst, the conversion of D-glucose increases to two times of that without a catalyst, accompanied by a slightly improved selectivity of fructose but substantially reduced selectivity towards sorbitol by ~2 times (Fig. 4d in Main text), illuminating that the MIL-100-Cr MOF can promote the D-glucose conversion and facilitate the formation of fructose, but restrain the sorbitol production.

For all DEMOfs, the selectivity to fructose is much higher while selectivity to sorbitol is lower compared with the pristine D₀. Upon increasing the incorporated amount of DL_x, the conversion of D-glucose and the selectivity to fructose show an overall stepwise increase, while the selectivity of sorbitol gradually decreases. When z reaches 50%, the D-glucose conversion is boosted about 2 times, and the selectivity to fructose increases by ~1.5 and 2 times for D_{1c} and D_{2c}, respectively. In contrast, the selectivity to sorbitol is reduced to 1/5 for both of them (Fig. 4d in Main text) in relation to that of the reaction over D₀. Evidently, both type-A and type-B defects can boost the activity of MIL-100-Cr to the D-glucose hydrogenation reaction and the selectivity to fructose, but reduce the selectivity to sorbitol. This is probably attributed to the weak coordination bonds between D-glucose and defective MSAS. Noticeably, type-A and type-B defects show distinguishing disparity on improvement of catalytic performances, which is mainly attributed to the hydrogen bonds between hydroxyl groups in glucose molecules and basic pyridyl-N atoms in DL₁ incorporated into the framework of Ru@D_{1a-c}. Compared with the blank reaction in absence of any catalysts, Ru NPs greatly increase the D-glucose

conversion by more than 10 times (Fig. 4d in Main text), but show a negligible effect on the selectivity to sorbitol and fructose. These results identified that the confined Ru NPs boost the reaction activity as they can decompose H₂ into active H atoms. In comparison with that of Ru NPs, parent MOF and DEMOF catalysts, the higher D-glucose conversion and sorbitol selectivity of Ru@D₀, Ru@D_{1a-c}, and Ru@D_{2a-c} can be attributed to the synergistic catalysis of the embedded small Ru particles, defect sites and pristine MSAS.

The above comprehensive results provide solid evidence for the following synergistic reaction mechanism of D-glucose selective hydrogenation to sorbitol catalyzed by Ru@DEMOFs (Fig. 1 in Main text): 1) H₂ molecules are first activated on the surface of Ru NPs and then dissociated into active H atoms to participate in the subsequent D-glucose hydrogenation reaction. 2) D-glucose molecules absorb on the surface of D_{1a-c} with type-A defects via coordination bonds between D-glucose and Cr-MSAS as well as hydrogen bonds between D-glucose and basic pyridyl-N atoms at defect sites, having decisive importance in controlling the speed of adsorption and desorption of reactants, intermediates and products on Ru@DEMOFs. 3) The activated H atoms transfer to D-glucose molecules with a subsequent hydrogenation to form sorbitol. 4) Finally, the target product sorbitol desorbs from the catalyst surface (Fig. 1f, g in Main text), and the Ru@DEMOFs catalysts are recovered to the original state for processing the next catalytic cycle. We note that the catalytic mechanism of Ru@D_{2a-c} is similar to that of Ru@D_{1a-c}, and the significant difference between them is that the absence of hydrogen bonds between D-glucose and the defect sites of DL2 in Ru@D_{2a-c} (Fig. 1g in Main text). The synergistic catalytic mechanism between Ru NPs, MSAS and DLx has been specified in the revised manuscript according to your valuable suggestions.

Main text Fig. 3i Routine FTIR spectra of the D-glucose impregnated G-Ru@D₀, G-Ru@D_{1c} and G-Ru@D_{2c}, in comparison with the activated unloaded Ru@D₀, Ru@D_{1c} and Ru@D_{2c} samples

Supplementary Fig. 32 Routine FTIR spectra of the D-glucose impregnated G-D₀, G-D_{1c} and G-D_{2c}, in comparison with the activated unloaded D₀, D_{1c} and @D_{2c} samples.

8. More reference about synergistic effect could be considered: for example, Adv Mater., 2019, 31,52,1904969 and Nature communications, 2021, 12(1): 1-9.

Answer: Thank you very much for your valuable suggestions. These recent excellent studies on the synergistic effect ^{3,4} are in line with our observation and thus we have cited them in the revised manuscript as a solid evidence (page 1, right column, line 9).

Reviewer #2 (Remarks to the Author):

The authors present a study of how defect engineering a MOF can have an influence on the incorporation of Ru nanoparticles and the consequent catalytic activity towards a hydrogenation reaction. A lot of high-quality work has been done here and there are some interesting conclusions about the influence on artificially-induced defects in MOFs. Whilst this has been reported by other researchers, I believe this is a noteworthy contribution to the body of literature.

Answer: We highly appreciate your positive evaluation of our work and are grateful for your constructive feedback that helps us improve the quality of the manuscript.

I have a few issues with some of the interpretation of the data, specifically:

1) P4 117 "The high quality UHV-FTIRS at ~110 K of Ru@D₀ show a main sharp peak at 2201 cm⁻¹ with a shoulder band peaked at 2194 cm⁻¹ stemming from CO-Cr³⁺ (Fig. 3a, b) and a weak band centered at 2153 cm⁻¹ assigned to CO adsorbed on the intrinsic defective Cr^{δ+}-CUSs (δ < 3) (Fig. 3e, f), in situ generated during synthesis." This assignment is made without any justification. The authors need to refer to previously published literature showing similar absorbances in a related system.

Answer: Thank you very much for your remark. The assignment of the Cr-related CO bands is justified by the related references (*J. Am. Chem. Soc.* 2006, 128, 3218-3227; *Chem. Soc. Rev.* 2010, 39, 4928-4950)^{5,6} in the revised manuscript (page 5, left column, line 5).

2) P5 113 "These results validate our proposal that the special type of artificially introduced defects have a unique regulation effect on the modality of Ru NPs." I don't understand what the authors mean by "regulation effect on the modality". Furthermore, the authors need to explain in the text above this sentence what the physical significance of the loss of spectral features in the FTIR spectra in Figures 3 e and f is.

Answer: We have followed your advice and the term "modality" has been corrected to "the morphology, electronic and steric properties" in the revised manuscript. We have re-analyzed the

spectral evolution at lower frequencies in the Ru-related CO vibration region with increasing the doping level of DL1 (Ru@D_{1a-c}, Fig. 3b in the revised version) and DL2 (Ru@D_{2a-c}, Fig. 3d in the revised version). The IR spectra are dominated by a rather broad feature ranging from 1980 to 2080 cm⁻¹, which varies in shape and intensity depending on the type and content of the incorporated DLs. These signals are characteristic for CO species bonded linearly to Ru atop sites. Fine structures are resolved only for Ru@D_{1c} with higher DL1 concentration (Fig. 3b in the revised version). Overall, the UHV-FTIRS data provide spectroscopic evidence for the presence of well-dispersed small Ru particles that feature a number of Ru sites with distinct steric (coordination) and electronic properties, thus leading to the overlapping of various CO bands bound to different Ru sites.

3) p6 line 5 "The higher stability of CO absorbed on the Ru NPs can be assigned to the reduced electron density of Ru NPs in Ru@D_{1a}, confirmed by XPS measurements (Supplementary Fig. 18)" The XPS in Fig. S18 DOES NOT support this conclusion. Their conclusion is based on the exact position of the Ru 3d⁵ peak which is barely discernible over the much stronger C1s peak in the spectra (particularly in subFigures a and b). Any measurement of peak position is therefore subject to massive uncertainty and consequently they can't draw any conclusions about charge transfer to/from the particles from this data. This analysis MUST be removed.

Answer: Thank you very much for drawing our attention to this important issue. Indeed, the binding energy of the 3d_{3/2} is very close to that of the more intense C1s peaks in the XPS spectra. However, the deconvoluted XPS data of Ru@D_{1a-c} and Ru@D_{2a-c} show a binding energy of Ru 3d_{5/2} at about 281 eV (Supplementary Fig. 11-12), which is clearly higher than that (~279.0 eV) observed for small neutral Ru particles, indicating strong electronic interactions between Ru particles and DEMOFs. This is further supported by XPS investigations of Au@DEMOFs as a reference system (Supplementary Fig. 31). We have followed your suggestions and removed the XPS data from the main text.

To provide deep insight into electronic structure of Ru NPs, we have reanalyzed the temperature-resolved UHV-FTIR data that allow us to gain more detailed insight into the active sites exposed by representative Ru@DEMOFs (Ru@D₀, Fig. 3f; Ru@D_{1a}, Fig. 3g; Ru@D_{2a}, Fig. 3h). Compared to Cr-related CO species, CO is more strongly bound to the confined Ru particles due to the

substantially enhanced electron π back-donation. Close inspections of the IR data show a higher thermal stability of CO species adsorbed on Ru@D_{1a} compared to Ru@D₀ and Ru@D_{2a} where the CO signals vanish nearly completely at about 450 K. This finding reveals that the interaction between CO and Ru NPs is enhanced by the type-A defects featuring basic pyridyl-N atoms. Despite the complex spectral pattern, two Ru-CO bands are resolved at 2016-2023 cm⁻¹ and 2070-2075 cm⁻¹ after heating the Ru@DEMOFs samples to 240 K (Fig. 3f-h). The occurrence of fine structures could be related to the thermal diffusion of CO to more stable sites, thus leading to an increase in ordering. The splitting of these two bands becomes slightly larger due to the modification of chemical environments along with the desorption of various CO species. The vibrational frequency and intensity ratio change depending on the type of DLx. The 2021 cm⁻¹ band is the predominant one for Ru@D₀ at 240 K, while the 2075 cm⁻¹ band becomes an intense one for Ru@D_{1a} and Ru@D_{2a}. This higher frequency vibration is characteristic for the more positively charged Ru sites, which could be formed via the electronic interaction between embedded Ru NPs and Cr ^{δ^+} -CUSs (Lewis acid sites) that lose one coordinating carboxylate. Furthermore, Ru@D_{1a} shows a lower population of the 2075 cm⁻¹ band compared with Ru@D_{2a} at higher temperatures, resulting from additional interaction between Ru particles and basic pyridyl-N atoms of DL1 that are absent in Ru@D_{2a}. Interestingly, a bridge Ru-related CO band is detected at 1883 cm⁻¹ as a weak signal for Ru@D_{2a} (Fig. 3h), revealing the presence of larger Ru NPs as minor species in Ru@D_{2a}, in line with the STEM observation (Fig. 2k).

We have added the temperature-dependent UHV-FTIRS data (Fig. 3f-h) and an intensive discussion in the revised manuscript to make this important point clear.

4) In summary, I believe this paper is suitable for publication in Nature Communications, subject to the above revisions being made.

Answer: Thank you very much for the positive judgement on the quality of our work. We have subjected the manuscript to a major revision in order to account for your concerns.

Reviewer #3 (Remarks to the Author):

In this work, the authors constructed a series of defect Ru@DEMOFs composite materials via introducing different amounts of isophthalic acid or 3,5-pyridinedipyridine in the MIL-100(Cr). Moreover, compared to the parent catalyst Ru@D₀, the defect Ru@DEMOFs has shown high catalytic performance and recoverability in the selective hydrogenation transformation of D-glucose. For further investigation, it is found that the synergistic effect of small defective molecules, Ru nanoparticles and metal nodes is the main factor to achieve the high catalytic performance. I think the authors have carefully characterized and discussed the defects. Unfortunately, from the perspective of the overall work, the current manuscript is not acceptable for Nature Communication. The reasons are listed as follow:

Answer: We highly appreciate your critical but constructive comments that help us improve the quality and raise the impact of our manuscript. We have followed your valuable suggestions and subjected the manuscript to a major revision in order to account for your concerns.

1. In the synthesis part, the mentioned method of generating defects via mixing the dicarboxylic acids in MOF has been widely reported. Therefore, the synthetic method in this work is lack of novelty.

Answer: We thank the reviewer for the positive feedback on the manuscript. Regarding the novelty of this work, we would like to point out that here, we do not deal with the exploitation of new method for artificially implanting defects but focus on solving the following key problems in MOF catalysis via defect engineering (DE) strategy:

1) It is well known that artificially implanted defects generally exacerbate the degradation of MOFs, leading to the poorer stability of resulting DEMOFs (see e.g. *Nat. Chem.*, 2019, 11, 622-628, citations 140; *Angew. Chem. Int. Ed.*, 2015, 54, 7234-7254, citations 627)^{7, 8}. In contrast, the present work proposed and demonstrated, for the first time, that the reactivity, selectivity and recyclability of Ru impregnated DEMOFs (Ru@DEMOFs) catalysts towards the biomass hydrogenation of glucose can be simultaneously boosted by rationally designing and precisely tuning defects. Our results subvert traditional cognition, and thus this work is of great importance and novelty in the development of defect engineering to exploit highly efficient recyclable MOF-based heterogeneous catalysts.

2) This work firstly discovered that the cationic framework of MIL-100-Cr exhibits different tolerances to various DLx based on solid and comprehensive experiment data.

3) Two different types of defects were designed and controllably implanted into the Ru NPs impregnated MIL-100-Cr (Ru@DEMOFs) via doping 3,5-pyridinedicarboxylate (DL1:PyDC²⁻, type-A defects) and m-phthalate (DL2:m-BDC²⁻, type-B defects), respectively. This work represents a significant conceptual advance in understanding the unique effect of each type of defects on the regulation of the amount, dispersion, size, shape and stabilization of small confined Ru NPs, and subsequently the activity, selectivity and recyclability of the resulting two types of Ru@DEMOFs.

4) Due to the complexity coming from the existence of counter anions, it remains obscure in the compensation way of the missing charges of the cationic framework, created via artificially implanted defects. The present work clarified that the missing charges were compensated by the modification of electronic and steric properties of framework metal coordination unsaturated sites, certified by the UHV-FTIRS using CO as probe molecule.

5) The evolution of defects during impregnating active moieties into DEMOFs has been demonstrated in this study, providing theoretical guidance for rationally adjusting the method of incorporation of active species to boost the catalytic performance of metal NPs imbedded MOFs.

6) The electron density of noble metal NPs and the adsorption mode of reactants play a crucial role in the hydrogenation activity of catalysts (see e.g. *Nature* 2016, 539, 76-80)⁹. Our work has firstly illuminated the special effect of each type of defects on the electronic properties of confined Ru NPs inside DEMOFs and the activation of reactants (D-glucose and hydrogen molecules).

7) This work fundamentally clarified the role of artificially implanted defects, impregnated active moieties and metal single active sites (MSAS) on the activity, selectivity and recyclability of Ru@DEMOFs. Based on the comprehensive and solid experimental data, we are able to propose a synergistic catalytic mechanism for the selective glucose hydrogenation reaction.

2. At the same time, the XRD results indicate that, after forming the defective MOF, the structure of MIL-100 has an apparent collapse, especially in the D_{1b-c} and the D_{2b-c} samples. I am afraid that the crystalline structures of defective MOFs with 30% or 50% doping dicarboxylic acid cannot be maintained. To make sure the MOF structure well remains, it is suggested to reduce the amount of

doping ligand. Moreover, the dicarboxylic acid amount in the defective MOF is not determined? This should be fixed.

Answer: The BET surface areas of $D_{1a'-c}$, $D_{2a'-c}$, $Ru@D_{1a'-c}$ and $Ru@D_{2a'-c}$ decrease along with incorporation of DLx ($x= 1, 2$), consistent with that of PCN-125 doped with the functionalized fragment of parent ligand TPTC (*J. Am. Chem. Soc.* 2012, 134, 20110)¹⁰. The slight loss of BET surface areas of $D_{1a'-b}$ and $D_{2a'-b}$ with significant increase of mesopores compared to D_0 demonstrates that $D_{1a'-b}$ and $D_{2a'-b}$ maintain the framework structure of MIL-100-Cr with good crystallinity as the presence of mesopores is typically accompanied with certain loss of specific surface areas for the modified MOFs (see e.g. *J. Am. Chem. Soc.* 2012, 134, 20110. *Chem. Mater.* 2010, 22, 4531. *Angew. Chem. Int. Ed.* 2008, 47, 9487. *J. Am. Chem. Soc.* 2012, 134, 126. *Angew. Chem. Int. Ed.* 2021, 60, 14601–14608, *Angew. Chem. Int. Ed.* 2015, 54, 13273 –13278, *Microporous Mesoporous Mater.* 2012, 154, 113)¹⁰⁻¹⁶. In comparison with $Ru@D_0$ ($1490.145 \text{ m}^2\text{g}^{-1}$), $Ru@D_{1b}$ and $Ru@D_{2a-b}$ show comparable BET values, ranging from 1404.548 to $1444.196 \text{ m}^2\text{g}^{-1}$, while $Ru@D_{1a'}$ ($1594.473 \text{ m}^2\text{g}^{-1}$), $Ru@D_{1a}$ ($1561.366 \text{ m}^2\text{g}^{-1}$) and $Ru@D_{2a'}$ ($1576.974 \text{ m}^2\text{g}^{-1}$) show substantially higher BET surface areas that decrease only slightly after loading Ru NPs. Moreover, mesopores are generated in $Ru@D_{1a-b}$ and $Ru@D_{2a-b}$. These results combined with the corresponding PXRD, FTIR, TGA results consistently confirm that $Ru@DEMOfs$ with low feeding ratios of DL to TLs ($z \leq 30\%$) retain the structural integrity of MIL-100 with good crystallinity.

The main peaks of D_{1c} , D_{2c} , $Ru@D_{1c}$ and $Ru@D_{2c}$ featuring the structure of MIL-100 are all present in their PXRD patterns (Supplementary Fig. 1), further confirming the maintenance of the crystalline frameworks. It is known that the smaller of the particle size, the broader of the bands in the PXRD patterns¹⁷⁻¹⁹. Therefore, the band broadening in the PXRD pattern of DEMOFs and $Ru@DEMOfs$ is attributed to the decrease of particle size for these crystalline power samples. This finding is supported by additional SEM images that show a particle size decreasing along with the DLs incorporation (Supplementary Fig. 33). Furthermore, the structure of MIL-100-Cr is also identified by the characteristic FTIR spectra (Supplementary Fig. 5). The TGA results reveal that D_{1c} , $Ru@D_{1c}$ and $Ru@D_{2c}$ are thermally more stable compared with D_0 , while D_{2c} shows a little bit decrease of thermal stability (Supplementary Fig. 3). Importantly, the BET surface areas of these samples remain high values ranging from 1027.526 to $1310.583 \text{ m}^2/\text{g}$ (Supplementary Fig. 16-21, Supplementary Tab. 4).

The reduced BET surface areas of D_{xc} ($x = 1, 2$) and $Ru@D_{xc}$ ($x = 1, 2$) compared with D_0 and $Ru@D_0$ don't result in the increase of mesopores, which could be explained by a certain degree of pore blocking due to the local disorder of these samples. Take D_{1c} as an example, its thermal decomposition after heating at 480 °C in N_2 leads to disappearance of almost all main peaks in PXRD (Response Fig. 1), and the profile of FTIR spectra is quite different from that of D_{1c} (Response Fig. 2). Simultaneously, the BET surface areas significantly decreased (Response Fig. 3). Most importantly, all these samples exhibit high D-glucose conversion ($\geq 99.0709\%$), selectivity to sorbitol ($\geq 86.9655\%$), and sorbitol yields ($\geq 86.2488\%$) for the first run (Supplementary Fig. 38) and excellent recyclability (can be recycled ≥ 9 runs). All these results definitely prove the maintenance of the framework structure of these samples.

Supplementary Fig. 33 SEM images of $Ru@D_0$ (a), $Ru@D_{1a}$ (b), $Ru@D_{1b}$ (c), and $Ru@D_{1c}$ (d) $Ru@D_{2a}$ (e), $Ru@D_{2b}$ (f), $Ru@D_{2c}$ (g).

Response Fig. 1 Powder X-ray diffraction patterns of D_{1c} before (a) and after calcination (b) under nitrogen at 480 °C.

Response Fig. 2 FTIR spectra of the dry, activated of D_{1c} before (a) and after calcination (b) under nitrogen at 480 °C.

Response Fig. 3 N₂ sorption isotherms at 78 K for D_{1c} before (a) and after calcination (b) under nitrogen at 480 °C.

In order to further find out the optimal feeding ratio of DLx to TL (z) ranging from 1% to 10%, we have conducted additional experiments and synthesized two types of DEMOFs and Ru@DEMOFs with z to be 5%. The corresponding results are shown in Supplementary Tab. 10 and Supplementary Fig.42 (please see below). The new results reveal that for the Ru@D_{1a'} and Ru@D_{2a'} DEMOF catalysts, the D-glucose conversion is comparable to that acquired for Ru@D₀, Ru@D_{1a} and Ru@D_{2a} DEMOFs, while the selectivity to sorbitol and the sorbitol yield are higher than that of Ru@D₀, but lower compared with Ru@D_{1a} and Ru@D_{2a}. The rate constant (*k*) and turnover frequency (TOF) of Ru@D_{1a'} and Ru@D_{2a'} are consistent with the expected values deduced from the evolution trends of *k* and TOF numbers upon the increase of incorporated defective ligands. A systematic comparison of D-glucose conversion (a), selectivity to sorbitol (b) and sorbitol yields (c) for D-glucose selective hydrogenation reaction catalyzed by these DEMOF samples for 12 recycles shows that the recyclability of Ru@D_{1a'} and Ru@D_{2a'} is higher than that of Ru@D₀ but lower compared with Ru@D_{1a} and Ru@D_{2a}. Overall, these results demonstrate that the optimal feeding molar ratio of DLs to total ligands for these two types of Ru-impregnated DEMOFs is 10%. The full set of data and detailed discussion on these new results have been added in Supplementary Information.

Supplementary Table 10. The list of D-glucose conversion, selectivity and yield of sorbitol, rate constant (k) and TOF for Ru@D₀, Ru@D_{1a'}, Ru@D_{1a}, Ru@D_{2a'} and Ru@D_{2a} toward the hydrogenation of D-glucose to sorbitol reaction for the first recycle.

	D-glucose conversion (%)	Selectivity (%)	Yield (%)	Rate constant (k)	Turnover frequency (TOF)
RuD ₀	99.071	89.826	88.991	0.0965	7415.2326
Ru@D _{1a'}	98.845	91.529	90.472	0.0237	1802.6575
Ru@D _{1a}	99.372	96.331	95.726	0.0246	2150.7299
Ru@D _{2a'}	99.283	92.318	91.657	0.0989	7495.4229
Ru@D _{2a}	99.343	93.417	92.803	0.1034	8201.4875

Supplementary Fig. 42 Comparison of D-glucose conversion (a), selectivity to sorbitol (b) and sorbitol yields (c) for hydrogenation of D-glucose to sorbitol reactions catalyzed by Ru@D₀, Ru@D_{1a'}, Ru@D_{1a}, Ru@D_{2a'} and Ru@D_{2a} for 12 recycles. Reaction conditions: glucose aqueous solution (25 wt%, 1.543 mol/L, 50 g), catalyst amount (1 g), H₂ (5.0 MPa), temperature (120 °C), stirring rate (600 rpm) and reaction time (150 minutes).

The amount of dicarboxylic acid in the DEMOF samples has been determined by the combination of element analysis and ¹H NMR. The corresponding results are displayed in Supplementary Fig. 7-10 and Supplementary Tables 2-3.

3. I found that the Ru nanoparticles are obtained via the impregnation method. This ship-in-a-bottle method results in the different size distributions of Ru NPs in each defective MOF, which has influence

on the electronic state of Ru NPs and the final catalytic performance. Therefore, control experiments should be carried out to exclude particle size effects during the reaction.

Answer: Thank you very much for this remark. We fully agree with you on this point. As you well know, the amount, dispersion, size, shape and stability of impregnated Ru NPs are highly dependent on the size, shape and chemical environment of pores as well as the porosity of materials due to the template effect of pores^{20, 21}. In this work, two types of DEMOFs are modified by the incorporation of DLx (x =1, 2) with different concentrations in a controlled way (Supplementary Tab. 6). The size, structure and performance of the confined Ru NPs in these Ru@D₀ and Ru@DEMOFs are synchronously regulated by the type and content of incorporated DLx. As regards the size distribution, we have carried out a systematic investigation of Ru@D₀ and various Ru@DEMOFs before and after 12 runs of catalysis using STEM (for details please see Fig. 2, Supplementary Fig. 25-28 and Tab. 6).

1.12 STEM analysis

Main Text Fig. 2 | Identification of Ru NPs in Ru impregnated parent and defect engineered MIL-100-Cr MOFs. a-c Schematic illustration of various Cr single active sites: **a** MSAS with type-A defect in D_{1a-c} and $Ru@D_{1a-c}$, formed via incorporation of DL1 with a weak ligator N at defect sites; **b** MSAS with type-B defect in D_{2a-c} and $Ru@D_{2a-c}$, generated by incorporation of DL2 without ligator at defect sites; **c** pristine Cr single active sites. **d-f** STEM images of **d** $Ru@D_{1a}$, **e** $Ru@D_{2a}$, and **f** $Ru@D_0$ before catalyzing biomass hydrogenation of D-glucose to sorbitol. **g-i** STEM images of **g** $Ru@D_{1a}$, **h** $Ru@D_{2a}$, and **i** $Ru@D_0$, after catalyzing biomass hydrogenation of glucose to sorbitol for 12 runs. **j-l** Size distributions of **j** $Ru@D_{1a}$, **k** $Ru@D_{2a}$, and **l** $Ru@D_0$ before (blue) and after (pink) 12 runs of catalysis.

Supplementary Fig. 25 STEM images of $Ru@D_{1b}$ (a), $Ru@D_{1c}$ (b), $Ru@D_{2b}$ (c), and $Ru@D_{2c}$ (d) before catalyzing biomass hydrogenation of D-glucose to sorbitol, and the STEM images of $Ru@D_{1b}$ (e), $Ru@D_{1c}$ (f), $Ru@D_{2b}$ (g), and $Ru@D_{2c}$ (h) after catalyzing biomass hydrogenation of D-glucose to sorbitol for 12 runs. Scale bars: 20 nm.

Supplementary Fig. 26 Particle size distributions of Ru NPs, decided from the STEM images, in Ru@D_{1b} (a), Ru@D_{1c} (b), Ru@D_{2b} (c), RuD_{2c} (d) before catalysis, and in Ru@D_{1b} (e), Ru@D_{1c} (f), Ru@D_{2b} (g), Ru@D_{2c} (h) after catalysis for 12 runs. Reaction conditions: D-glucose aqueous solution (25 wt%, 1.543 mol/L, 50 g), catalyst amount (1 g), H₂ (5.0 MPa), temperature (120 °C), and stirring rate (600 rpm).

Supplementary Fig. 27 STEM images of Ru@D_{1a'} (a) and Ru@D_{2a'} (b). Scale bars: 20 nm.

Supplementary Fig. 28 Particle size distributions of Ru NPs, decided from the STEM images, in Ru@D_{1a'} (a), Ru@D_{2a'} (b).

The STEM images (Supplementary Fig. 27) and the statistics of particle size distribution of Ru NPs (Supplementary Fig. 28) show that the size evolution of confined Ru NPs in these DEMOFs

with feeding ratio (z) ranging from 0% to 10% is consistent with that of Ru@DEMOFs with higher content of DLx (Supplementary Fig. 25-26).

Supplementary Tab. 6 Comparison of the sizes of dominate Ru NPs in Ru@DEMOFs before and after 12 runs of hydrogenation reaction.

Catalyst name	Dominate Ru NPs diameter before 12 runs (nm)	Dominate Ru NPs diameter after 12 runs (nm)	Diameters increase ratio (%) of dominate Ru NPs diameter
Ru@D ₀	1.98	2.45	23.73
Ru@D _{1a'}	1.81	/	/
Ru@D _{1a}	1.95	2.20	12.83
Ru@D _{1b}	2.05	2.48	20.97
Ru@D _{1c}	2.85	4.77	67.36
Ru@D _{2a'}	1.98	/	/
Ru@D _{2a}	1.93	2.20	13.99
Ru@D _{2b}	2.11	2.57	21.80
Ru@D _{2c}	2.72	5.80	113.23

Particle size distributions of Ru NPs in all measured samples, determined by the STEM images, show that the diameters of dominant Ru NPs in both two kinds of Ru@DEMOFs increase along with the feeding ratio of DLx (x = 1, 2) to TL. These results illuminate that the Ru NPs sizes can be controllably adjusted by the introduced defects of different concentrations. After 12 runs of D-glucose selective hydrogenation, the size of dominant Ru NPs is enlarged (Fig. 2g-l; Supplementary Fig. 25, 26), and the higher concentration of defects leads to the larger aggregation degree of Ru NPs in these

two types of Ru impregnated DEMOFs. However, the aggregation degree of Ru NPs in both Ru@D_{1a-b} and Ru@D_{2a-b} is lower than that in Ru@D₀ (Fig. 2g-i; Supplementary Tab. 6), demonstrating both types of defects of low concentration can stabilize Ru NPs. Furthermore, the aggregation degree of Ru NPs in Ru@D_{1a-c} is lower than that in Ru@D_{2a-c} with the same z of DLx, illuminating that the type-A defects can stabilize Ru NPs more efficiently than type-B defects against aggregation during catalytic reaction, mainly due to the stronger anchoring effect between confined Ru NPs and basic pyridyl-N atoms at type-A defects. The above results demonstrate that a rational tuning of defects can prevent the aggregation of Ru NPs embedded in Ru@DEMOFs.

Finally, we would like to note that the emphasis of this work is on a fundamental understanding of the unique effect of different types of defects on regulating the amount, dispersion, size, shape and stability of the confined Ru NPs in Ru@DEMOFs and thus on the catalytic performance of these Ru@DEMOFs catalysts towards selective hydrogenation of D-glucose to sorbitol. Again, we agree with you that it would be interesting to investigate particle size effects in more detail. However, we feel that it is beyond the scope of this paper and we will focus on this topic in a future work.

4. The authors mentioned that the surface electronic state of Ru nanoparticles is changed in the defective MOFs, but how electron transfer affects the catalytic performance has not been verified. This is an important factor for the catalytic performance – it is necessary to demonstrate this point. In addition, it is assumed that the presence of H-bond in the defect MOF (D_{1a-c}) has interaction with the substrate leading to the high catalytic performance. I think it is necessary to conduct the control experiments and provide related evidence to confirm this point. Moreover, a reliable reaction mechanism of D-glucose transformation in this work is expected.

Answer: Thank you for your valuable suggestions. To provide deep insight into electronic structure and size of Ru NPs, we have reanalyzed the temperature-resolved UHV-FTIR data that allow us to gain more detailed insight into the active sites exposed by representative Ru@DEMOFs (Ru@D₀, Fig. 3f; Ru@D_{1a}, Fig. 3g; Ru@D_{2a}, Fig. 3h). Compared to Cr-related CO species, CO is more strongly bound to the confined Ru particles due to the substantially enhanced electron π back-donation. Close inspections of the IR data show a higher thermal stability of CO species adsorbed on Ru@D_{1a} compared to Ru@D₀ and Ru@D_{2a} where the CO signals vanish nearly completely at about 450 K.

This finding reveals that the interaction between CO and Ru NPs is enhanced by the type-A defects featuring basic pyridyl-N atoms. Despite the complex spectral pattern, two Ru-CO bands are resolved at 2016-2023 cm^{-1} and 2070-2075 cm^{-1} after heating the Ru@DEMOFs samples to 240 K (Fig. 3f-h). The occurrence of fine structures could be related to the thermal diffusion of CO to more stable sites, thus leading to an increase in ordering. The splitting of these two bands becomes slightly larger due to the modification of chemical environments along with the desorption of various CO species. The vibrational frequency and intensity ratio change depending on the type of DLx. The 2021 cm^{-1} band is the predominant one for Ru@D₀ at 240 K, while the 2075 cm^{-1} band becomes an intense one for Ru@D_{1a} and Ru@D_{2a}. This higher frequency vibration is characteristic for the more positively charged Ru sites, which could be formed via the electronic interaction between embedded Ru NPs and Cr^{δ+}-CUSs (Lewis acid sites) that lose one coordinating carboxylate. Furthermore, Ru@D_{1a} shows a lower population of the 2075 cm^{-1} band compared with Ru@D_{2a} at higher temperatures, resulting from additional interaction between Ru particles and basic pyridyl-N atoms of DL1 that are absent in Ru@D_{2a}. Interestingly, a bridge Ru-related CO band is detected at 1883 cm^{-1} as a weak signal for Ru@D_{2a} (Fig. 3h), revealing the presence of larger Ru NPs as minor species in Ru@D_{2a}, in line with the STEM observation (Fig. 2k).

The XPS data provide further evidence for the electronic structure modification of Ru particles. The deconvoluted XPS data of Ru@D_{1a-c} and Ru@D_{2a-c} show a binding energy of Ru 3d_{5/2} at about 281 eV (Supplementary Fig. 11-12), which is higher than that observed for small neutral Ru particles (~279.0 eV)²², indicating strong electronic interactions between Ru particles and DEMOFs. These IR and XPS results validate that the special type of artificially introduced defects has a unique regulation effect on the morphology, electronic and steric properties of the embedded Ru particles. This is further supported by XPS investigations of Au@DEMOFs as a reference system (Supplementary Fig. 31).

Supplementary Fig. 31 XPS spectra of Au 4f_{7/2}/4f_{5/2} doublet region obtained for Au@D₀, Au@D_{1c} and Au@D_{2c}.

We have conducted additional experiments in order to gain detailed information about the unique effects of each type of implanted defects on the adsorption mode of D-glucose reactant and thus to further investigate the synergistic catalytic mechanism between Ru NPs, MSAS, and DLx towards the D-glucose hydrogenation reaction. Firstly, we have systematically synthesized a series of D-glucose impregnated DEMOF samples including D₀ (G-D₀), D_{1c} (G-D_{1c}), D_{2c} (G-D_{2c}), Ru@D₀ (G-Ru@D₀), Ru@D_{1c} (G-Ru@D_{1c}) and Ru@D_{2c} (G-Ru@D_{2c}). In the next step, these samples were characterized by FTIR spectroscopy, and the corresponding results are shown in Fig. 3i in the main text and Supplementary Fig. 32. Clearly, the broad band centered at ~3384 cm⁻¹, assigned to the -OH stretching mode of D-glucose molecules in G-D₀ and G-Ru@D₀, shifts to lower frequency after doping with defective linkers (DL1, DL2), demonstrating the presence of weak coordination bonds between D-glucose and defective MSAS. Interestingly, the -OH vibrations of G-D_{1c} and G-Ru@D_{1c} shows a further redshift in frequency compared to those of G-D_{2c} and G-Ru@D_{2c}, revealing the formation of hydrogen bonds between hydroxyl groups in glucose molecules and basic pyridyl-N atoms in DL1 incorporated into the framework of Ru@D_{1c}. Moreover, the disparity of OH-bands between the G-DEMOFs and G-Ru@DEMOFs further indicates that the Ru NPs impregnation process results in the evolution of both types of defects. Overall, these results validate our proposed synergistic catalytic

mechanism of D-glucose selective hydrogenation to sorbitol for the two different kinds of Ru-NPs impregnated DEMOFs (Fig. 1 in Main text).

To get an in-depth understanding of the reaction mechanism, the role of pristine MSAS, artificially implanted defects and Ru NPs as well as their synergetic effect on the high conversion (> 98%) of D-glucose and selectivity to sorbitol (> 85%) during the catalytic process have been investigated. Without any catalyst, the conversion of D-glucose is as low as 4.22%, accompanied by a 57.45% selectivity to sorbitol and 16.91% selectivity to mannitol (Fig. 4d in Main text). When D₀ containing a little amount of intrinsic defects (Fig. 3d in Main text) is used as catalyst, the conversion of D-glucose increases to two times of that without a catalyst, accompanied by a slightly improved selectivity of fructose but substantially reduced selectivity towards sorbitol by ~2 times (Fig. 4d in Main text), illuminating that the MIL-100-Cr MOF can promote the D-glucose conversion and facilitate the formation of fructose, but restrain the sorbitol production.

For all DEMOFs, the selectivity to fructose is much higher while selectivity to sorbitol is lower compared with the pristine D₀. Upon increasing the incorporated amount of DLx, the conversion of D-glucose and the selectivity to fructose show an overall stepwise increase, while the selectivity of sorbitol gradually decreases. When z reaches 50%, the D-glucose conversion is boosted about 2 times, and the selectivity to fructose increases by ~1.5 and 2 times for D_{1c} and D_{2c}, respectively. In contrast, the selectivity to sorbitol is reduced to 1/5 for both of them (Fig. 4d in Main text) in relation to that of the reaction over D₀. Evidently, both type-A and type-B defects can boost the activity of MIL-100-Cr to the D-glucose hydrogenation reaction and the selectivity to fructose, but reduce the selectivity to sorbitol. This is probably attributed to the weak coordination bonds between D-glucose and defective MSAS. Noticeably, type-A and type-B defects show distinguishing disparity on improvement of catalytic performances, which is mainly attributed to the hydrogen bonds between hydroxyl groups in glucose molecules and basic pyridyl-N atoms in DL1 incorporated into the framework of Ru@D_{1a-c}. Compared with the blank reaction in absence of any catalysts, Ru NPs greatly increase the D-glucose conversion by more than 10 times (Fig. 4d in Main text), but show a negligible effect on the selectivity to sorbitol and fructose. These results identified that the confined Ru NPs boost the reaction activity as they can decompose H₂ into active H atoms. In comparison with that of Ru NPs, parent MOF and

DEMOF catalysts, the higher D-glucose conversion and sorbitol selectivity of Ru@D₀, Ru@D_{1a-c}, and Ru@D_{2a-c} can be attributed to the synergistic catalysis of the embedded small Ru particles, defect sites and pristine MSAS.

The above comprehensive results provide solid evidence for the following synergistic reaction mechanism of D-glucose selective hydrogenation to sorbitol catalyzed by Ru@DEMOFs (Fig. 1 in Main text): 1) H₂ molecules are first activated on the surface of Ru NPs and then dissociated into active H atoms to participate in the subsequent D-glucose hydrogenation reaction. 2) D-glucose molecules absorb on the surface of D1a-c with type-A defects via coordination bonds between D-glucose and Cr-MSAS as well as hydrogen bonds between D-glucose and basic pyridyl-N atoms at defect sites, having decisive importance in controlling the speed of adsorption and desorption of reactants, intermediates and products on Ru@DEMOFs. 3) The activated H atoms transfer to D-glucose molecules with a subsequent hydrogenation to form sorbitol. 4) Finally, the target product sorbitol desorbs from the catalyst surface (Fig. 1f, g in Main text), and the Ru@DEMOFs catalysts are recovered to the original state for processing the next catalytic cycle. We note that the catalytic mechanism of Ru@D_{2a-c} is similar to that of Ru@D_{1a-c}, and the significant difference between them is that the absence of hydrogen bonds between D-glucose and the defect sites of DL2 in Ru@D_{2a-c} (Fig. 1g in Main text).

We have added the temperature-dependent UHV-FTIRS data of various Ru@DEMOFs (Fig. 1f-h in main text) and the IR spectra of D-glucose impregnated DEMOFs and Ru@DEMOFs (Fig. 1i in main text and Supplementary Fig. 32) in the revised manuscript. We have followed your valuable suggestions and discussed these important issues (electronic properties of the embedded Ru particles, synergistic catalytic mechanism between Ru NPs, MSAS, and DLx) in more detail.

Main text Fig. 3i Routine FTIR spectra of the D-glucose impregnated G-Ru@D₀, G-Ru@D_{1c} and G-Ru@D_{2c}, in comparison with the activated unloaded Ru@D₀, Ru@D_{1c} and Ru@D_{2c} samples.

Supplementary Fig. 3j Routine FTIR spectra of the D-glucose impregnated G-D₀, G-D_{1c} and G-D_{2c}, in comparison with the activated unloaded D₀, D_{1c} and @D_{2c} samples.

5. In the recyclability test, please give the TEM results of Ru@D₀ after 12 runs of reaction.

Answer: The STEM images of Ru@D₀ after 12 runs of reaction have been shown in Fig. 2e of the original version and in Fig. 2i of the revised manuscript.

Reference:

1. Feng, X. et al. Engineering a highly defective stable UiO-66 with tunable lewis-bronsted acidity: The role of the hemilabile linker. *J. Am. Chem. Soc.* **142**, 3174-3183 (2020).
2. Cai, G. & Jiang, H. L. A modulator-induced defect-formation strategy to hierarchically porous metal-organic frameworks with high stability. *Angew. Chem. Int. Ed.* **56**, 563-567 (2017).
3. Chang, G. G. et al. Construction of hierarchical metal-organic frameworks by competitive coordination strategy for highly efficient CO₂ conversion. *Adv. Mater.* **31**, 1904969 (2019).
4. Liu, Y. et al. Fabricating polyoxometalates-stabilized single-atom site catalysts in confined space with enhanced activity for alkynes diboration. *Nat. Commun.* **12**, 4205 (2021).
5. Vimont, A. et al. Investigation of acid sites in a zeotypic giant pores chromium(III) carboxylate. *J. Am. Chem. Soc.* **128**, 3218-3227 (2006).
6. Vimont, A., Thibault-Starzyk, F. & Daturi, M. Analysing and understanding the active site by IR spectroscopy. *Chem. Soc. Rev.* **39**, 4928-4950 (2010).
7. Fang, Z. et al. Defect-engineered metal-organic frameworks. *Angew. Chem. Int. Ed.* **54**, 7234-7254 (2015).
8. Liu, L. et al. Imaging defects and their evolution in a metal-organic framework at sub-unit-cell resolution. *Nat. Chem.* **11**, 622-628 (2019).
9. Zhao, M. et al. Metal-organic frameworks as selectivity regulators for hydrogenation reactions. *Nature* **539**, 76-80 (2016).
10. Park, J. et al. Introduction of functionalized mesopores to metal-organic frameworks via metal-ligand-fragment coassembly. *J. Am. Chem. Soc.* **134**, 20110-20116 (2012).
11. Albolqany, M. K. et al. Molecular surgery at microporous MOF for mesopore generation and renovation. *Angew. Chem. Int. Ed.* **60**, 14601-14608 (2021).
12. Kim, Y. et al. Hydrolytic transformation of microporous metal-organic frameworks to hierarchical micro- and mesoporous MOFs. *Angew. Chem. Int. Ed.* **54**, 13273-13278 (2015).
13. Qiu, L. G. et al. Hierarchically micro- and mesoporous metal-organic frameworks with tunable porosity. *Angew. Chem. Int. Ed.* **47**, 9487-9491 (2008).
14. Diring, S. et al. Controlled multiscale synthesis of porous coordination polymer in nano/micro regimes. *Chem. Mater.* **22**, 4531-4538 (2010).
15. Sun, L. B. et al. Cooperative template-directed assembly of mesoporous metal-organic frameworks. *J. Am. Chem. Soc.* **134**, 126-129 (2012).
16. Klimakow, M. et al. Characterization of mechanochemically synthesized MOFs. *Microporous Mesoporous Mater.* **154**, 113-118 (2012).
17. Scherrer, P. Göttinger Nachrichten Math. (1918).
18. Warren, B. E. X-ray Diffraction. (Dover publicatioin; 1969).
19. Weibel, A. et al. The big problem of small particles: A comparison of methods for determination of particle size in nanocrystalline anatase powders. *Chem. Mater.* **17**, 2378-2385 (2005).
20. Goswami, S. et al. Pore-templated growth of catalytically active gold nanoparticles within a metal-organic framework. *Chem. Mater.* **31**, 1485-1490 (2019).
21. Mian, M. R. et al. Precise control of Cu nanoparticle size and catalytic activity through pore templating in Zr metal-organic frameworks. *Chem. Mater.* **32**, 3078-3086 (2020).
22. Elmasides, C. et al. XPS and FTIR study of Ru/Al₂O₃ and Ru/TiO₂ catalysts: reduction characteristics and interaction with a methane-oxygen mixture. *J. Phys. Chem. B* **103**, 5227-5239 (1999).

REVIEWER COMMENTS

Reviewer #1 (Remarks to the Author):

The authors have modified the manuscript properly and replied the comments in detail that I think it could be considered to accepted.

Reviewer #2 (Remarks to the Author):

The authors have addressed my concerns for the most part - however i must still take issue with the XPS analysis presented in figures S 11 and S 12. They infer charge transfer effects "electrons transfer from

Ru NPs to the framework of MIL-100-Cr DEMOF" based on the position of a deconvoluted Ru3d5 peak. As i previously mentioned, this peak position is uncertain at best as in some of the subfigures (S11 a+b and S12 a+b) the fitted peak for Ru3d it is so small that it is highly questionable if it even exists (the overall fit would not look any different if that peak were removed).

Moreover, whilst I accept that for the stronger Ru peaks shown (fig S11 c+d and fig S12 c) they are at higher binding energies than references for bulk Ru metal, this cannot simply and conclusively be ascribed to charge transfer. There are multiple other phenomena which would also cause a binding energy shift such as oxidation (Ru(IV) species are reported at ~280.5-281 eV which would fit their peak position) or size effects (it has been well documented that small nanoparticles of metals exhibit anomalously high XPS binding energies even the absence of charge-transfer effects.

The authors need to remove the subfigures where the peak fitting is questionable (11 a+b and 12 a+b) and they need to at least discuss the other two effects that would cause high binding energy of their Ru 3d5 peaks (oxidation and size effect) and attempt to motivate why they think charge transfer is the only possible explanation.

Reviewer #3 (Remarks to the Author):

The authors have well dealt with the questions and I think the current manuscript can be accepted.

Dear reviewers,

Thank you very much for your positive feedback and constructive comments that have improved the clarity and raised the impact of our manuscript. We sincerely hope that we have adequately accounted for all your concerns by more discussion and explanation. All changes in the main text are highlighted in blue, and the point-by-point responses to all your comments are listed below:

Reviewer #1 (Remarks to the Author):

The authors have modified the manuscript properly and replied the comments in detail that I think it could be considered to accepted.

Answer: We highly appreciate your positive evaluation of the modified manuscript.

Reviewer #2 (Remarks to the Author):

The authors have addressed my concerns for the most part - however I must still take issue with the XPS analysis presented in Figures S 11 and S 12. They infer charge transfer effects "electrons transfer from Ru NPs to the framework of MIL-100-Cr DEMOF" based on the position of a deconvoluted Ru 3d₅ peak. As I previously mentioned, this peak position is uncertain at best as in some of the subfigures (S11 a+b and S12 a+b) the fitted peak for Ru3d it is so small that it is highly questionable if it even exists (the overall fit would not look any different if that peak were removed). Moreover, whilst I accept that for the stronger Ru peaks shown (Fig S11 c+d and Fig S12 c) they are at higher binding energies than references for bulk Ru metal, this cannot simply and conclusively be ascribed to charge transfer. There are multiple other phenomena which would also cause a binding energy shift such as oxidation (Ru(IV) species are reported at ~280.5-281 eV which would fit their peak position) or size effects (it has been well documented that small nanoparticles of metals exhibit anomalously high XPS binding energies even the absence of charge-transfer effects).

The authors need to remove the subfigures where the peak fitting is questionable (11 a+b and 12 a+b) and they need to at least discuss the other two effects that would cause high binding energy of their Ru 3d₅ peaks (oxidation and size effect) and attempt to motivate why they think charge transfer

is the only possible explanation.

Answer: Thank you very much for this remark. We fully agree with your points that the band of Ru 3d is overlapped by the more intense C1s peaks in the XPS spectra, and the oxidation and size effects could also influence the binding energy of Ru 3d peaks. Given that all Ru impregnated MOF catalysts Ru@D₀, Ru@D_{1a-c} and Ru@D_{2a-c} were obtained at the same reduction condition (under 4 MPa hydrogen pressure at 120 °C for 180 min) in a stainless autoclave, the effect of oxidized Ru species on the binding energy of Ru 3d can be excluded. Considering that the temperature-resolved UHV-FTIR data has already provided deep insight into electronic structure and size of Ru NPs, while the binding energy of Ru 3d is obtained by fitting these bands, we have removed Figure S11 and Figure S12 along with their discussions, according to your valuable suggestions.

Reviewer #3 (Remarks to the Author):

The authors have well dealt with the questions and I think the current manuscript can be accepted.

Answer: We highly appreciate your positive evaluation of the modified manuscript.

REVIEWER COMMENTS

Reviewer #2 (Remarks to the Author):

I'm satisfied that my concerns have been addressed. The manuscript can now be accepted.

Reviewer #2 (Remarks to the Author):

I'm satisfied that my concerns have been addressed. The manuscript can now be accepted

Answer: Thank you for your positive evaluation of the modified manuscript.